# Comprehensive assessment of physiological responses in women during the ESA dry immersion VIVALDI microgravity simulation

Adrien Robin [1] ✉, Angelique Van Ombergen[2], Claire Laurens [3], Audrey Bergouignan[4], Laurence Vico [5], Marie-Thérèse Linossier[5], Anne Pavy-Le Traon[6], Marc Kermorgant [6], Angèle Chopard[7], Guillaume Py[7], David Andrew Green[8], Michael Tipton [9], Alexander Choukér[10], Pierre Denise [11], Hervé Normand[11], Stéphane Blanc[12], Chantal Simon[13], Elisabeth Rosnet[14], Françoise Larcher[15], Peter Fernandez[5], Isabelle de Glisezinski[3], Dominique Larrouy[3], Isabelle Harant-Farrugia[3], Inês Antunes[16], Guillemette Gauquelin-Koch[17], Marie-Pierre Bareille[18], Rebecca Billette De Villemeur[18], Marc-Antoine Custaud [1] ✉ & Nastassia Navasiolava [1] ✉

Astronauts in microgravity experience multi-system deconditioning, impacting their inflight efficiency and inducing dysfunctions upon return to Earth gravity. To fill the sex gap of knowledge in the health impact of spaceflights, we simulate microgravity with a 5-day dry immersion in 18 healthy women (ClinicalTrials.gov Identifier: NCT05043974). Here we show that dry immersion rapidly induces a sedentarily-like metabolism shift mimicking the beginning of a metabolic syndrome with a drop in glucose tolerance, an increase in the atherogenic index of plasma, and an impaired lipid profile. Bone remodeling markers suggest a decreased bone formation coupled with an increased bone resorption. Fluid shifts and muscular unloading participate to a marked cardiovascular and sensorimotor deconditioning with decreased orthostatic tolerance, aerobic capacity, and postural balance. Collected datasets provide a comprehensive multi-systemic assessment of dry immersion effects in women and pave the way for future sex-based evaluations of countermeasures.

Spaceflights have shown the possibilities and limitations of adaptation to space since the first human microgravity exposure experienced by Yuri Gagarin in 1961. For the last 60 years, results have confirmed that the space environment, and microgravity in particular, cause physiological multisystem deconditioning which may impact astronauts' inflight efficiency, and create difficulties upon their return to gravity[1]. These physiological changes under weightlessness are now better understood and reflect the body's adaptation to a new environment: significant loss of bone, loss of muscle mass and strength, hormonal and metabolic changes, cardiovascular and sensorimotor deconditioning, circadian rhythms, immune, and even persistent ophthalmologic changes now defined by NASA as spaceflight-associated neuro-ocular syndrome (SANS)[2].

Currently, females represent only 12% of humans who have crossed the one-hundred km high Karman line separating Earth's atmosphere and outer space, with 78 out of 628 total space travelers being women (as of December 2022). However, it is likely that future exploration crews will be mixed[3], including for the lunar surface

missions and beyond, as mixed group dynamics appear to be facilitated[4]. To ensure health and safety during deep space missions, it is imperative to consider the influence of sex on flight-induced physiological and psychological changes, both in terms of deconditioning and the efficiency of countermeasures. Reviews by Mark et al. and Evans et al. have demonstrated some sex-related differences in cardiovascular, immunologic, sensorimotor, musculoskeletal, reproductive, and behavioral adaptations to spaceflight. It appears that women are less susceptible to SANS, however, they have long been known to be more susceptible to post-flight orthostatic intolerance, along with having more pronounced hypovolemia and baroreflex impairment, and lower vasoconstrictive reserve[5,6]. Moreover, artificial gravity countermeasure for cardiovascular deconditioning may be less efficient in women[3,6], though it remains unclear if this is also true for other organ systems. There may also be differences in hormone, stress, and immune responses, the sensory system, and the circadian system; understanding these differences is important for planning missions and designing spacecrafts. However, due to the low number of female astronauts it is difficult to reach conclusions; it is unclear whether these are individual or sex differences. Moreover, few ground-based simulations have included female participants. The WISE-2005 study (Women's International Space Simulation for Exploration, sponsored by NASA, the Canadian Space Agency, the French National Space Agency, and the European Space Agency - ESA) is currently the only prolonged simulation study (60-day Head Down Bed Rest - HDBR) performed specifically in women[7].

Understanding the underlying mechanisms of inter-system processes of adaptation and deconditioning, and enhancement of countermeasures, remain a challenge and major priority for manned space programs. Thus, space agencies are actively engaged in studying the physiological adaptations to the space environment through studies onboard the International Space Station (ISS) but, due to the limited number of manned flight opportunities and the complexity of inflight experiments, ground-based simulations are also being employed. During the early 1970s, an era of active development of space programs, the Soviets proposed the dry immersion (DI) method to simulate the effects of prolonged microgravity. The DI model provides a unique opportunity to study the physiological effects of the lack of a supporting structure for the body[8]. Exploring the nature of DI in women in a high-fidelity environment can uniquely provide an integrative and more holistic view of the physiology of human adaptation to microgravity. Because of the potential benefits for spaceflight, as well as for research towards mitigating the health consequences in sedentary populations and bedridden patients, ESA has decided to include this model in its research program. As a first step, ESA decided to carry out standardization work like that done on the bedrest model aiming to expand the current knowledge and harmonize protocols.

Until 2020, only male volunteers took part in DI studies, primarily because mainly men participated in actual spaceflights, but also because urination in the DI bath is more complicated for women. A first pilot 3-day non-strict DI study in 6 healthy women was conducted in Russia in September–November 2020[9–11], using a special urine collector. It yielded novel data and demonstrated feasibility, tolerance, and the absence of substantial medical risks for DI with female participants.

The present study falls within this context and is the first strict DI without periods of sitting or standing carried out in Europe in women. Its objective was a comprehensive and integrative assessment of the physiological changes induced by 5 days of strict DI in females, to provide a control dataset for further DI studies examining the efficacy of countermeasures. The main physiological systems were explored before, during, and after the 5 days of immersion through a comprehensive set of tests and measurements. Results were analyzed by a multidisciplinary expert group to better understand the physiological changes elicited by exposure to DI and to compare the responses with those observed during bedrest studies and spaceflight.

## Results
A total of 18 female participants (29 ± 1 yr; see Supplementary Table 1 for baseline characteristics) entered the 5-day DI study (see Fig. 1 for DI model representation and study summary). All 18 participants completed the entire protocol−4 Baseline days (B4 to B1), 5 days of DI (D1 to D5), and 2 Recovery days (R0, R1).

### Feasibility and adjustments for DI on women
The 5-day DI without authorized periods of sitting or standing was proved feasible in women. No major medical issues or adverse events were reported. For daily shower and weighing, participants were transferred to a specific platform maintained at −6° head-down position. To limit the discontinuation of immersion for urination and perform urine collection, Purewick™ system was available. This system, primarily designed for women suffering from urinary incontinence, aspirates urine from a soft flexible wick (external catheter) to a sealed collector. Participants positioned the system by themselves before urination, then removed until the next urination. Also, bed-side urinary basin for urination in supine position out-of-bath on lifting platform was available. The preferences and requests of the participants were respected. Three participants used only Purewick™, three used both options (mostly Purewick™), and twelve used only basin.

### Global tolerance and out-of-bath time
The study was well tolerated by participants. Blood assessment performed by MEDES at B4 and R1 for medical safety reasons showed no pathological values (Supplementary Table 2). On the first night, general discomfort expectedly increased from 1 to 5 (out of 10), and back pain from near 0 to 4−5 points, whereas sleep quality decreased from 8 to 5 and was negatively associated with the severity of back pain (Pearson $r = -0.63$; $P = 0.005$). Symptoms and complaints progressively decreased in the following days of DI (Supplementary Fig. 1a). We observed important inter-participant variability in perceived comfort. Fluid shift-related complaints remained at non-significant level, except for nausea in one participant (Supplementary Fig. 1b). Variations in blood biology remained within normal ranges throughout the study.

Participants were lifted out of water for daily hygiene and some out-of-bath tests (leg plethysmography, bio-impedance, magnetic resonance imaging (MRI), and dual-energy x-ray absorptiometry (DXA), mostly during the last two days of immersion). Out-of-bath time was about 10 h for a total of 120 h, and no >3 h per day (Supplementary Fig. 1c), which is comparable to the usual out-of-bath time for DI protocols. For example, out-of-bath time during the 5-day DI protocol conducted in 18 men (MEDES, France, 2018–2019) was $9.7 ± 1.3\,h$[12]. Urination option had only minor effect on total out-of-bath time, which consisted 9 h 16 min for Purewick only, 9 h 19 min for combined, and 10 h 47 min for basin only (Supplementary Fig. 1d).

Complete test schedules are presented in Fig. 1d.

### DI induces thoraco-cephalic fluid shift with early hypovolemia through hormonal regulation
Astronauts in space undergo relative hypovolemia, starting early following microgravity exposure due to the thoraco-cephalic fluid shift, which also occurs during DI. Here, as partial water balance decreased during DI of about 0.5 L.day⁻¹ (Fig. 2a; Supplementary Table 3), we looked at the main hormones involved in blood volume regulation. The brain natriuretic peptide (BNP) is released by cardiomyocytes in response to stretching upon an increase in ventricular pressure. The renin is secreted by the kidneys and participates in increasing the extracellular fluid volume (e.g., plasma volume, PV). Here we observed a significant increase in BNP (+111 ± 23%) concomitant with a decrease

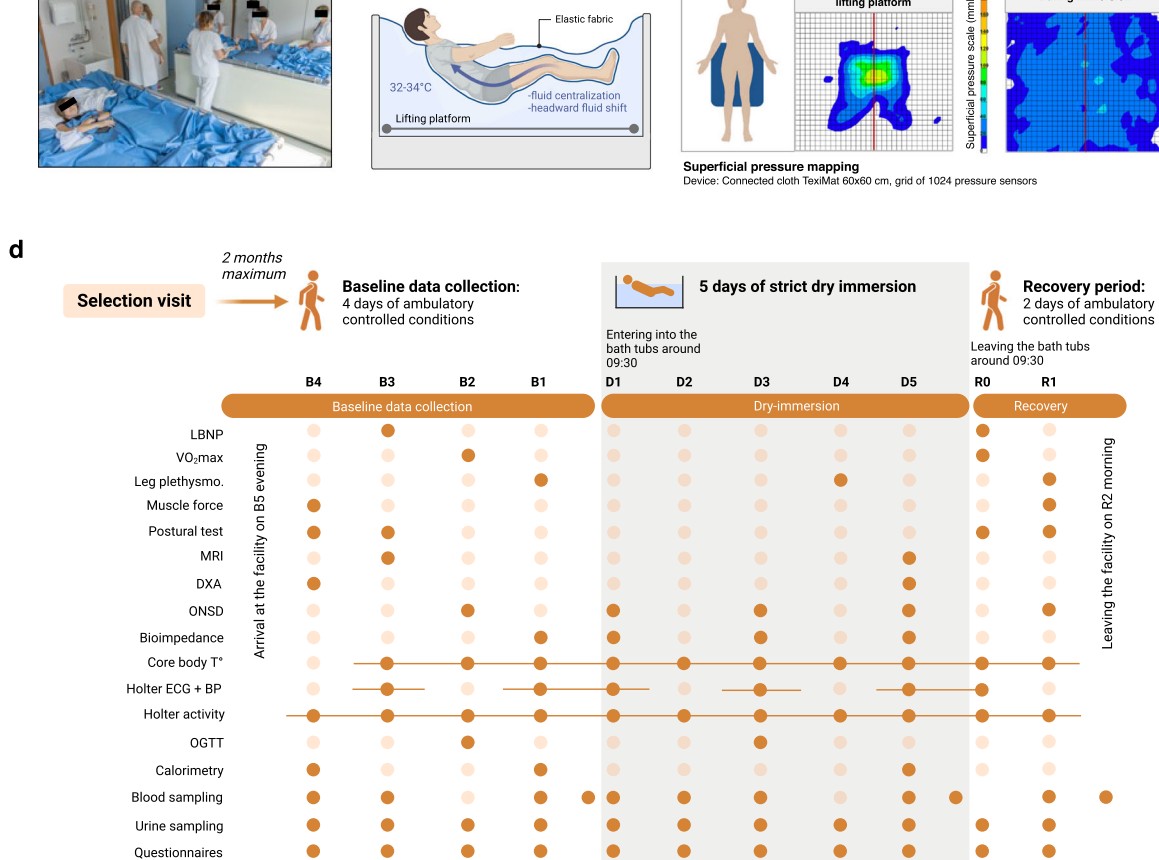

**Fig. 1 | The DI model in MEDES space clinic (Toulouse, France). a** Two participants in two separate baths underwent strict DI simultaneously. **b**, **c** The DI model simulates the lack of support and unloading observed in microgravity, as shown by the superficial pressure mapping where the pressure applied by the body is evenly distributed over the sensors during bath immersion. **d** Operational scenario and main testing protocols are represented throughout the study period. The participants arrived at the facility in the evening of B5, four days of baseline data collection have been performed (B4–B1). The participants entered the bathtubs at D1 -09:30 and left the bath at the same hour at R0 (five full days of DI: D1–D5). Two days of recovery data collection were performed (R0–R1). Participants left the facilities in the morning of R2. The tests performed are represented by a filled dot, and a solid line indicates a continuous recording.

in renin (−55 ± 8%) and aldosterone (−34 ± 6%) during the first 8 h following the onset of immersion. These early acute changes were followed by resetting the next days with a decrease in BNP and an increase in renin and aldosterone compared to B1 (−62 ± 8% for BNP, +301 ± 66% for renin, +204 ± 52% for aldosterone at D5) (Fig. 2d; Supplementary Table 4).

We collected urine samples every day and assessed urine electrolytes. Natriuresis and, to a lesser extent, kaliuresis, increased on the first day of DI, then decreased at D3, while free water clearance did not differ from baseline (Fig. 2b; Supplementary Table 3; Supplementary Fig. 2). However, blood electrolytes ($Na^+$, $Cl^-$, $K^+$), as well as blood osmolality, remained within normal range (Supplementary Fig. 2; Supplementary Table 4).

We assessed the PV evolution estimated by Hb·Hct Dill & Costill formula (−12 ± 2% at the evening of first DI day followed by a stabilization at −18-24% till the end of DI, with −24 ± 1% at D3, −21 ± 2% at D5, and −19 ± 2% at R0) (Fig. 2c) and by proteinemia (−4 ± 1% at the evening of first DI day, −9 ± 1% at D3, and −7 ± 1% at D5), the difference between methods suggesting partial blood protein transfer to the interstitial sector. Along with the decreased water balance, the total body water (TBW) decreased during DI (−4.0 ± 0.4% at D5), mostly due to extracellular fluid loss (Fig. 2e). This reduction in fluid content led to a −6% loss in arm, leg, and calf segmental volumes, and up to −12% loss for the torso (Fig. 2e). Taken together, these data indicate reduced overall fluid content and hypovolemia, starting in the first hours of DI and stabilized during the following days of DI. This closely mimics the hypovolemia state reported during spaceflights[13].

PV depletion correlated with body height (taller participants lost more in PV), orthostatic tolerance (OT) loss (more hypovolemic subjects decreased more in OT-index), and back pain (maximal reported back pain was higher in more hypovolemic subjects).

Astronauts suffer from SANS related to the thoraco-cephalic fluid shift. Given that we observed fluid changes in DI, we also examined Optic Nerve Sheath Diameter (ONSD) ($n = 17$; ONSD was not clearly defined in 1 participant). Compared to baseline, ONSD increased significantly by +7 ± 2% at D1, +10 ± 2% at D3, +14 ± 1% at D5 and remained +6 ± 2% higher at R1 (Fig. 2f). Increase in ONSD at the end of DI (D5) correlated with PV diminution at the beginning of DI (D1 evening) (Pearson $r = 0.51$, $P = 0.036$): those who lost PV faster, has a greater increase in ONSD. Also, it correlated with maximal back pain (Pearson $r = −0.54$, $P = 0.03$): those who experienced more pain had a greater increase in ONSD.

## DI induces acute cardiovascular and sensorimotor deconditioning

The cardiovascular deconditioning in astronauts returning to Earth is characterized by an orthostatic intolerance, a decrease in aerobic capacity, and an increased heart rate. To evaluate the DI effect on OT, we performed lower body negative pressure (LBNP) test. Data were analyzed for 17 subjects (one participant (H) presented acute

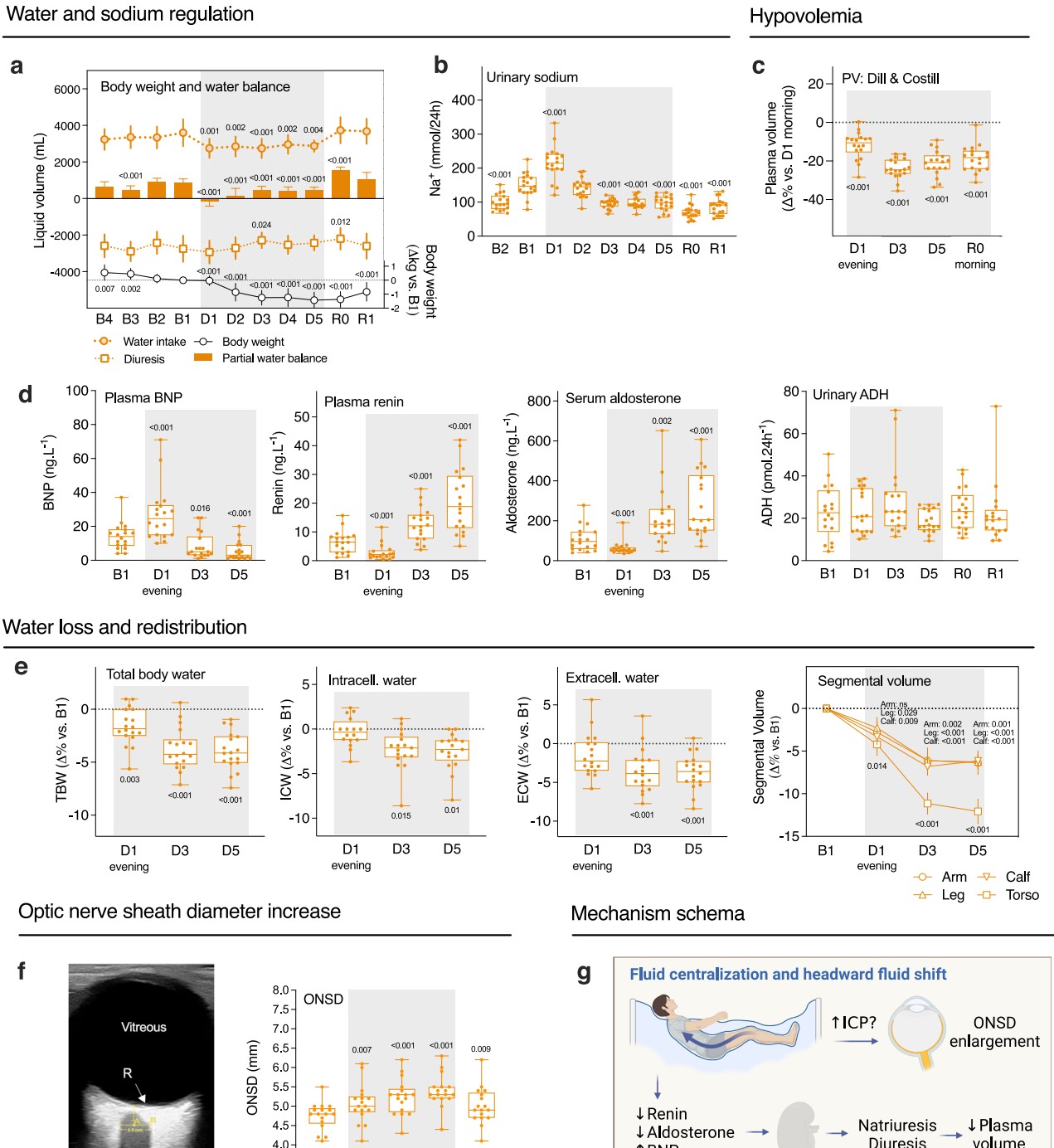

**Fig. 2 | The thoraco-cephalic fluid shift rapidly induces hypovolemia, water loss and redistribution, and ONSD increase. a** Body weight evolution (right axis), and partial water balance calculated from water intake and diuresis (left axis) vs. B1 ($n = 18$). **b** Time course of natriuresis vs. B1 ($n = 18$). **c** Plasma volume variations vs. D1 morning estimated from hematocrit-hemoglobin count (Dill and Costill method; PV ($\Delta\%$) = 100 × [HbB(1−0.01Hcti)]/[Hbi(1−0.01HctB)]−100, where HbB and HctB are baseline Hb and Hct levels, and Hbi and Hcti are Hb and Hct on further days). **d** Time course of variations in hormones (BNP, renin, aldosterone, ADH) involved in the fluid regulation loop, vs. B1 ($n = 18$). **e** Fluid compartments (total, intracellular, and extracellular water) and segmental volumes variations estimated by bio-impedance, compared to B1 morning ($n = 18$). **f** Ocular ultrasonography (R retina, ON optic nerve) of ONSD measured 3 mm behind the posterior globe vs. B2 ($n = 17$). **g** Schema summarizing the DI effects on ONSD and

volemia. Box plots indicate minimum, 25th percentile, median, 75th percentile, and maximum values. Statistical significance was determined compared with baseline by one-way RM ANOVA test, with post-hoc Bonferroni multiple-comparison performed if global ANOVA P value < 0.05. Global ANOVA results: **a** Body weight $P < 0.001$, partial water balance $P < 0.001$, water intake $P < 0.001$, diuresis $P < 0.001$. **b** Urinary sodium $P < 0.001$. **c** Plasma volume $P < 0.001$. **d** BNP $P < 0.001$, renin $P < 0.001$, aldosterone $P < 0.001$, ADH $P = 0.157$. **e** TBW $P < 0.001$, ICW $P = 0.002$, ECW $P < 0.001$, segmental volumes $P < 0.001$. **f** ONSD $P < 0.001$. Shading indicates DI period. **a** Data are mean ± sd. **e** For segmental volume, data mean ± s.e.m. ADH antidiuretic hormone, BNP brain natriuretic peptide, ECW extracellular body water, ICP intracranial pressure, ICW intracellular body water, ONSD optic nerve sheath diameter, PV plasma volume, TBW total body water.

LBNP test: number of finishers at each step and orthostatic tolerance index

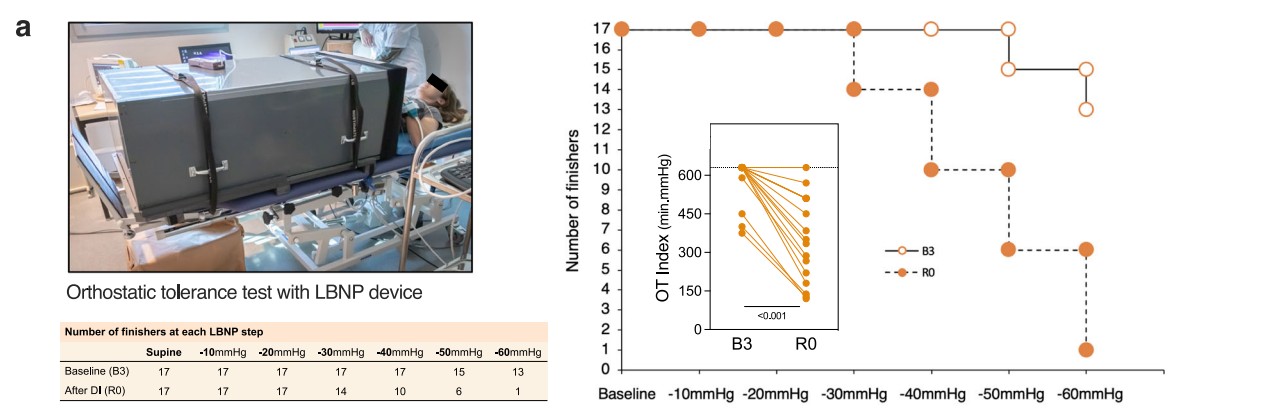

Aerobic capacity test: V̇O₂max, maximum load, heart rate, RER and core body temperature

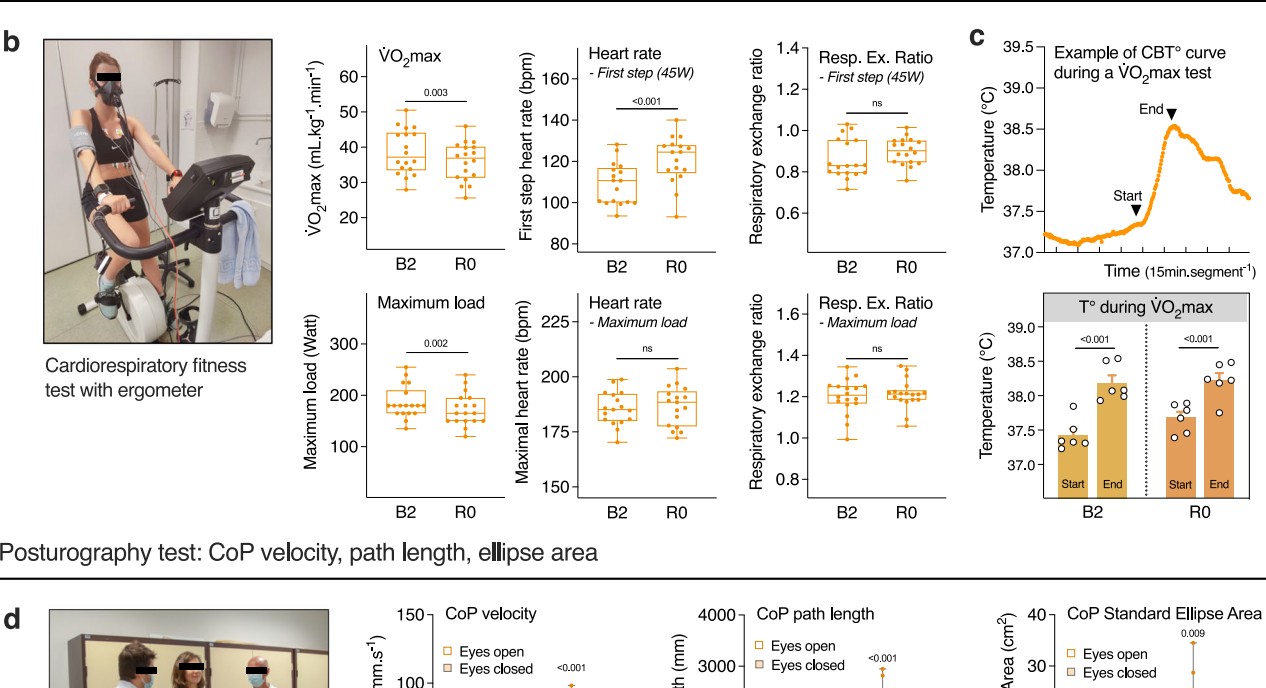

Posturography test: CoP velocity, path length, ellipse area

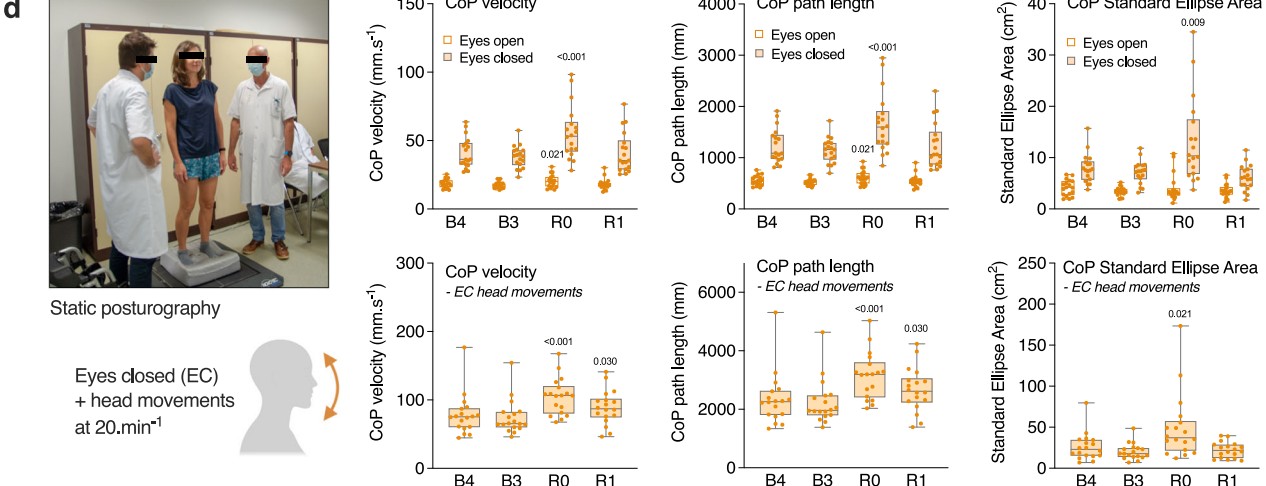

bradycardia at the last step of pre-DI LBNP test, and was not tested post-DI for safety reasons). We observed pronounced decrease in tolerance to LBNP after DI, with tolerance time drop from $17.3 \pm 0.4$ min at B3 to $12.8 \pm 0.9$ min at R0.

Tolerance for different LBNP steps is shown in Fig. 3a. Before DI all 17 subjects finished 40 mmHg LBNP step; 13 subjects accomplished the totality of 6 steps. Immediately after DI, 7 subjects were intolerant to 40 mmHg step; only 1 subject finished the totality of six steps. We observed a decrease in OT index of $-41 \pm 6\%$ ($-230 \pm 31$ min mmHg) (Fig. 3a). Diminution in OT-index at R0 correlated with height (Pearson $r = 0.57$, $P = 0.017$, taller participants lost more in OT to LBNP), PV diminution at D5 (Pearson $r = 0.57$, $P = 0.016$,

**Fig. 3 | DI induces acute cardiovascular and sensorimotor deconditioning.**
**a** Lower Body Negative Pressure test (LBNP). Table and Kaplan-Meier survival graph show the number of test finishers at each LBNP step before (B3; white circles) and after DI (R0; filled circles). Individual orthostatic tolerance (OT) index was calculated as the sum of the product of duration (min completed for each LBNP step) and negative pressure (mmHg) at each completely or partially accomplished LBNP step (*n* = 17). **b** Cardiorespiratory fitness assessed by the maximal volume of oxygen uptake per minute (V̇O$_{2max}$), maximal load developed, first-step, and maximal heart rate before (B2) and after DI (R0). The respiratory exchange ratio (RER) is calculated as the V̇CO$_2$ production divided by the V̇O$_2$ consumption (*n* = 18). **c** Core body temperature recorded during the aerobic fitness test, and the Start temperature compared to the End one (*n* = 6). **d** Static posturographic parameters (center of pressure velocity, path length, and standard ellipse area) with head fixed + eyes open, head fixed + eyes closed, and with dynamic head movements + eyes closed. CoP center of pressure, EC eyes closed (*n* = 18). Box plots indicate minimum, 25th percentile, median, 75th percentile, and maximum values. Bars represent mean ± s.e.m. **a**, **b** Two-tailed paired Student *t* test. **c** Two-way RM ANOVA with post hoc Bonferroni multiple comparisons; with immersion effect *P* = 0.215 and test effect *P* < 0.001. **d** One-way RM ANOVA with post hoc Bonferroni multiple comparisons compared with B3. Global ANOVA results: CoP velocity eyes open *P* = 0.021 and eyes closed *P* < 0.001 and with head movements *P* < 0.01, CoP path length eyes open *P* = 0.021 and eyes closed *P* < 0.001 and with head movements *P* < 0.001, CoP SEA eyes open *P* = 0.271 and eyes closed *P* < 0.001 and with head movements *P* = 0.008. CoP center of pressure, RER respiratory exchange ratio, SEA standard ellipse area.

those who lost more PV lost more in OT), maximal voluntary contraction of knee extensors (Pearson *r* = 0.54, *P* = 0.03, those who lost more in force lost more in OT) and maximal back pain (Pearson *r* = −0.6, *P* = 0.009, those who experienced more pain lost more in OT). However, there was no correlation with a change in maximal oxygen uptake (V̇O$_{2max}$).

To evaluate the maximal aerobic capacity, a V̇O$_{2max}$ test was performed on a cycle ergometer in the upright seated position, before (B2) and after the DI period (R0). There was a decrease in V̇O$_{2max}$ from 38.7 ± 1.5 to 35.9 ± 1.3 mL kg$^{-1}$ min$^{-1}$ (−6.7 ± 1.9%) and maximum load developed (−7.3 ± 2.1%) after DI, and an increase in first-step heart rate (HR) of +12 ± 2 bpm (from 108 ± 3 bpm—at B2 to 121 ± 3 bpm—7 hours after the end of DI), suggesting a post-DI tachycardia (Fig. 3b). Six participants out of 18 preserved pre-DI V̇O$_{2max}$ level. Change in V̇O$_{2max}$ tended to be associated with initial fitness at baseline (*r* = −0.39, *P* = 0.12; more fitted tended to lose more in V̇O$_{2max}$), but was not associated with changes in PV or OT-index. Finally, no significant effect of DI has been shown for core body temperature increase during the V̇O$_{2max}$ test (temperature maximum 38.1 ± 0.1 °C at B2 vs. 38.3 ± 0.1 at R0) (Fig. 3c). Taken together, these results demonstrate a cardiovascular deconditioning induced by DI.

We observed an impaired sensorimotor function at R0, as shown by decrease in postural stability (Fig. 3d), especially with eyes closed. Increase in center of pressure (CoP) velocity and CoP path length vs. B3 baseline consisted +17 ± 5% for eyes open, +45 ± 9% for eyes closed, and +46 ± 8% for eyes closed with head movements. Increase in CoP standard ellipse area was of +16 ± 13% for eyes open, +83 ± 23% for eyes closed, and +188 ± 72% for eyes closed with head movements. At R1, posturographic impairment remained statistically significant only for the most discriminative condition, eyes closed with head movements (increase in CoP velocity and path length of +24 ± 8%).

## DI induces lower limb deconditioning

As a result of fluid shifts, astronauts exhibit thinner legs and facial swelling, known as bird legs and puffy face[14]. We performed MRI in both legs before (B3) and at the end of DI (D5) to assess calf muscle and skin volume evolution. Decrease of −6 ± 1% was observed for calf muscle cross-section area (CSA), and −5 ± 1%—for calf skin CSA (Fig. 4a). We also performed a DXA scan to explore lean mass variations in the lower limbs area. Lean mass was reduced (−5 ± 1%) for both legs on the last day of DI (D5) compared to baseline (B4) (Fig. 4b).

Loss in muscle mass, tone and blood volume during spaceflight lead to increased venous compliance. Calf plethysmography was performed to estimate venous compliance (Fig. 4c). Contrary to spaceflight, we observed a significant decrease in venous compliance during DI (−32 ± 7% at D4), returning to the baseline value after one day of recovery (−5 ± 5% at R1).

To assess muscle force, we performed isometric contractions for main leg muscle groups (knee and ankle flexors and extensors) before

DI and after one day of recovery (*n* = 17) (Fig. 4d). The maximal isometric torque significantly decreased for knee extensors (quadriceps) from 133 ± 7 Nm at B4 to 123 ± 6 Nm at R1 (−8 ± 2%), and was not significantly modified for other muscle groups.

## DI drastically decreases activity, dampens circadian variations of heart rate and blood pressure, and lowers core body temperature

Astronauts in space are no longer carrying their own body weight. Their postural muscles required on Earth to maintain vertical posture are completely discharged, and movements require less efforts. As a result, they are exposed to a decreased physical activity mimicking a sedentarily deconditioning behavior. We estimated physical activity using ankle (*n* = 17) and wrist (*n* = 16) accelerometers. DI induced an acute decrease in daily activity counts, especially at the leg level (Fig. 5a). Summarized counts at ankle for daytime consisted 1.287 ± 0.072 × 10$^6$ at baseline (average for B4-B1), vs. 0.136 ± 0.014 × 10$^6$ under immersion (average for D2-D5), revealing 10-times decrease. At wrist, summarized counts for daytime were 1.730 ± 0.088 × 10$^6$ at baseline vs. 1.154 ± 0.072 × 10$^6$ under immersion, with only 1.5-times decrease.

Night-time activity was significantly lower than daytime activity, with an average of 0.035 ± 0.006 × 10$^6$ summarized counts for the ankle and 0.076 ± 0.010 × 10$^6$ for the wrist at baseline. During DI, these values decreased to 0.016 ± 0.002 × 10$^6$ for ankle and increased to 0.112 ± 0.015 × 10$^6$ for wrist (non-significantly). This resulted in about 40-fold day-night difference for the ankle and a 20-fold difference for the wrist at baseline, while under DI this difference decreased to about 10-fold for both the ankle and the wrist (Fig. 5a).

To assess the effects of simulated weightlessness on the cardiovascular circadian rhythm, we recorded continuous heart rate and blood pressure (BP) with the SOMNOtouch™ device. HR, SBP, and DBP data averaged over 15 min are plotted in Fig. 5b for each day of recording (B3, B1, D1, D3, and D5). We also calculated daytime (07:00–22:59) and nighttime (23:00–06:59) mean values and the difference between them to evaluate the circadian variations (Fig. 5c). HR slightly decreased under DI. SBP and DBP slightly increased beginning with B1 baseline; this increase lasted till the end of DI. We observed a decrease in the day-night variations of HR at D3 compared to B3 (from 16 ± 1 to 11 ± 1 bpm, *P* < 0.05; Fig. 5b), as well as a decrease in both systolic (−5 mmHg vs B3, *P* < 0.05) and diastolic (−4 mmHg vs B3, *P* < 0.05) blood pressure day-night variations during DI (Fig. 5c). Nocturnal dipping at D3 compared to B3 decreased from 21 ± 1% to 15 ± 1% for HR, from 10 ± 1% to 5 ± 1% for SBP, and from 13 ± 1% to 6 ± 1% for DBP.

During spaceflight, thermoregulation can become critical with a risk of hyperthermia, as convective and evaporative heat losses are impaired in space due to gravity absence[15]. Moreover, the homeostatic set-point of core temperature itself may be modified, and some astronauts report thermal discomfort. Here we continuously recorded core body temperature with ingestible pill telemetry. Temperature

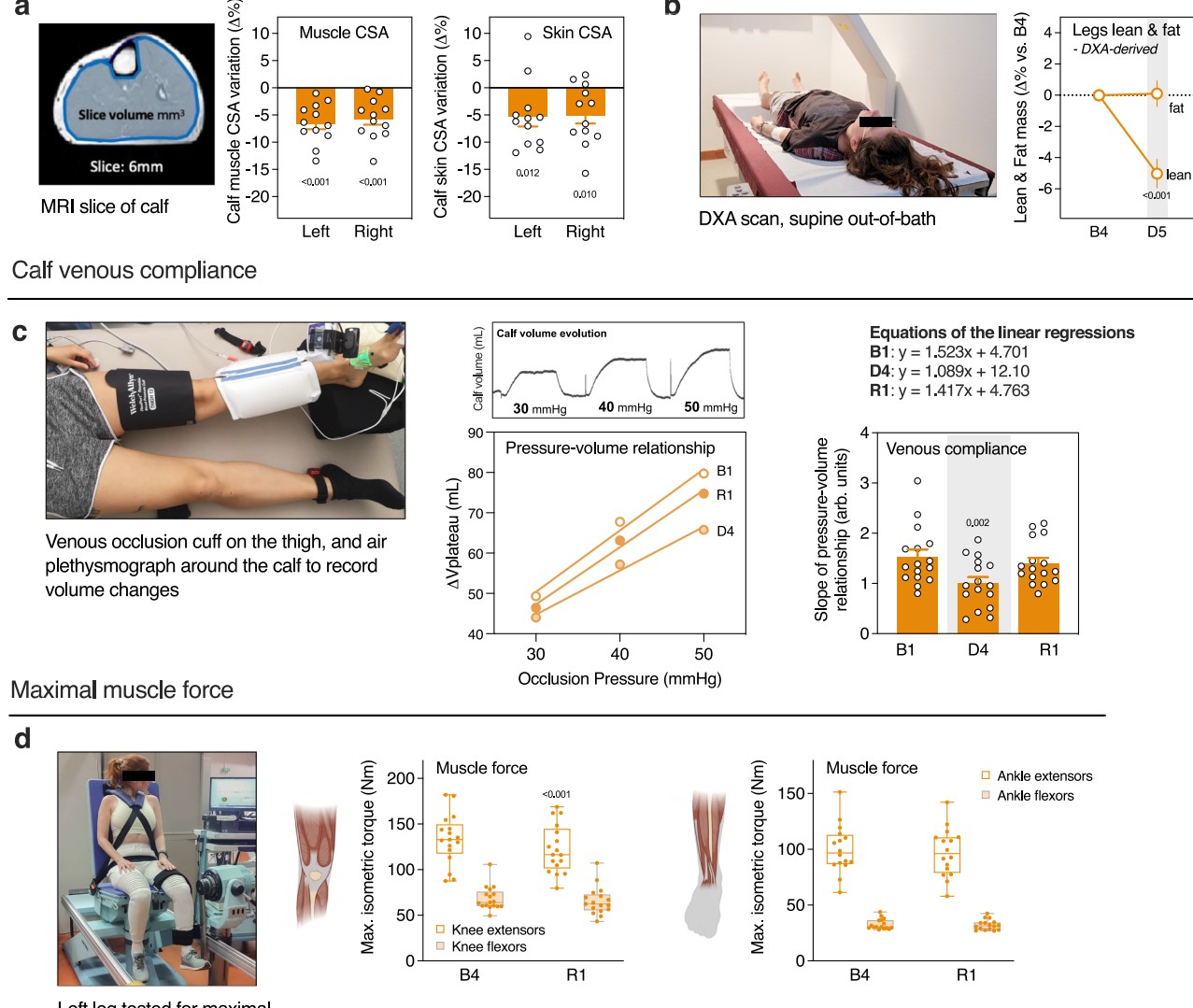

Calf muscle and skin cross-section area, and legs composition

Calf venous compliance

Maximal muscle force

**Fig. 4 | DI induces lower limbs volume and lean mass loss, and muscles weakening. a** Calf muscles and skin CSA evolution assessed by MRI (*n* = 12) and calculated using the largest slices of both left and right calves. **b** DXA-derived legs lean and fat mass (*n* = 18). **c** Calf venous compliance assessed by air plethysmography. We performed 30, 40, 50 mmHg occlusion steps. The calf volume evolution is plotted against occlusion pressure to obtain pressure-volume relationships. The slopes of the regression lines of these relationships were used as measure of venous compliance. The left leg was tested (*n* = 16). **d** Maximal isometric torque for knee and ankle extensors and flexors during voluntary contraction before (B4) and one day after DI (R1). The left leg was tested (*n* = 17). Box plots indicate minimum, 25th percentile, median, 75th percentile, and maximum values. Bars represent mean ± s.e.m. **a, b, d** Two-tailed paired Student's *t* test. **c** One-way RM ANOVA with post-hoc Bonferroni multiple comparisons compared with B1, Global ANOVA results: venous compliance *P* < 0.001. CSA cross-section area.

data averaged over 15 min are plotted in Fig. 5d. The baseline curve represents the mean for days from B3 to B1, during DI−from D1 to D5, and recovery−from R0 to R1. We also calculated a daytime (07:00–22:59; 37.18 ± 0.04 °C at B vs. 37.08 ± 0.07 at DI vs. 37.31 ± 0.05 at R), a nighttime (23:00–06:59; 36.72 ± 0.07 at B vs. 36.57 ± 0.08 at DI vs. 36.81 ± 0.07 at R), and a mesor (24 h) mean temperature (37.02 ± 0.05 at B vs. 36.90 ± 0.07 at DI vs. 37.15 ± 0.06 at R) for each day (Fig. 5e). During DI, core body temperature decreased of about 0.10−0.15 °C, while circadian rhythms were globally preserved.

**Short-term DI induces loss of lean body mass and metabolic alterations**
A significant decrease in body weight of 1.5−2 kg was observed after 5 days of DI (Fig. 2a) due to loss of lean body mass documented by both

bio-impedance analysis (BIA) (−1.7 ± 0.2 kg, *P* < 0.001) and DXA (−1.8 ± 0.1 kg, *P* < 0.001) methods (Fig. 6a), going along with extracellular water (ECW) loss (Fig. 2e) and probably reflecting mostly muscle dehydration. DXA-derived fat mass varied without significance (−0.1 ± 0.1 kg), whereas BIA-derived fat mass showed a gain of +0.3 ± 0.1 kg (main DI effect *P* = 0.012) (Fig. 6a). The latter might indicate a positive energy balance, i.e., energy intakes were superior to the actual energy needs[16]. Resting metabolic rate and non-protein respiratory quotient (npRQ) were unchanged (Fig. 6c) indicating no alteration in energy metabolism and in the proportion of carbohydrate over fats being used as fuel. Regarding protein metabolism, urinary nitrogen increased at D5 (+18 ± 4%, *P* < 0.001) (Fig. 7c; Supplementary Table 3) indicating a greater disappearance of protein as fuel and protein catabolism. Plasma and urine creatinine did not

### Physical activity continuous recording

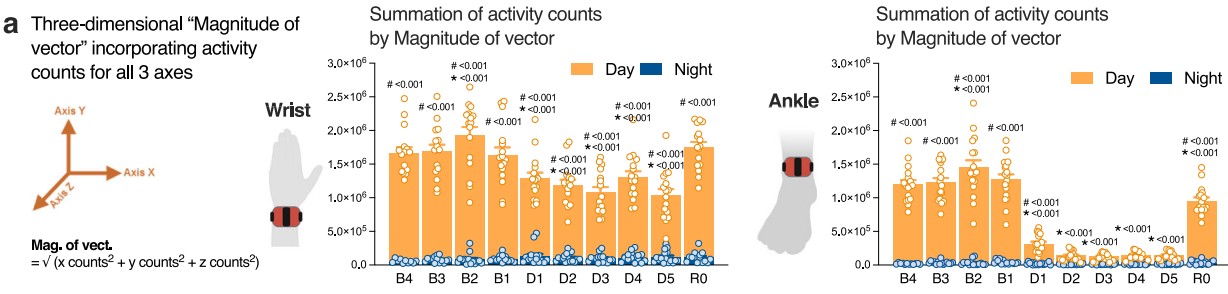

### Heart rate and blood pressure continuous recording

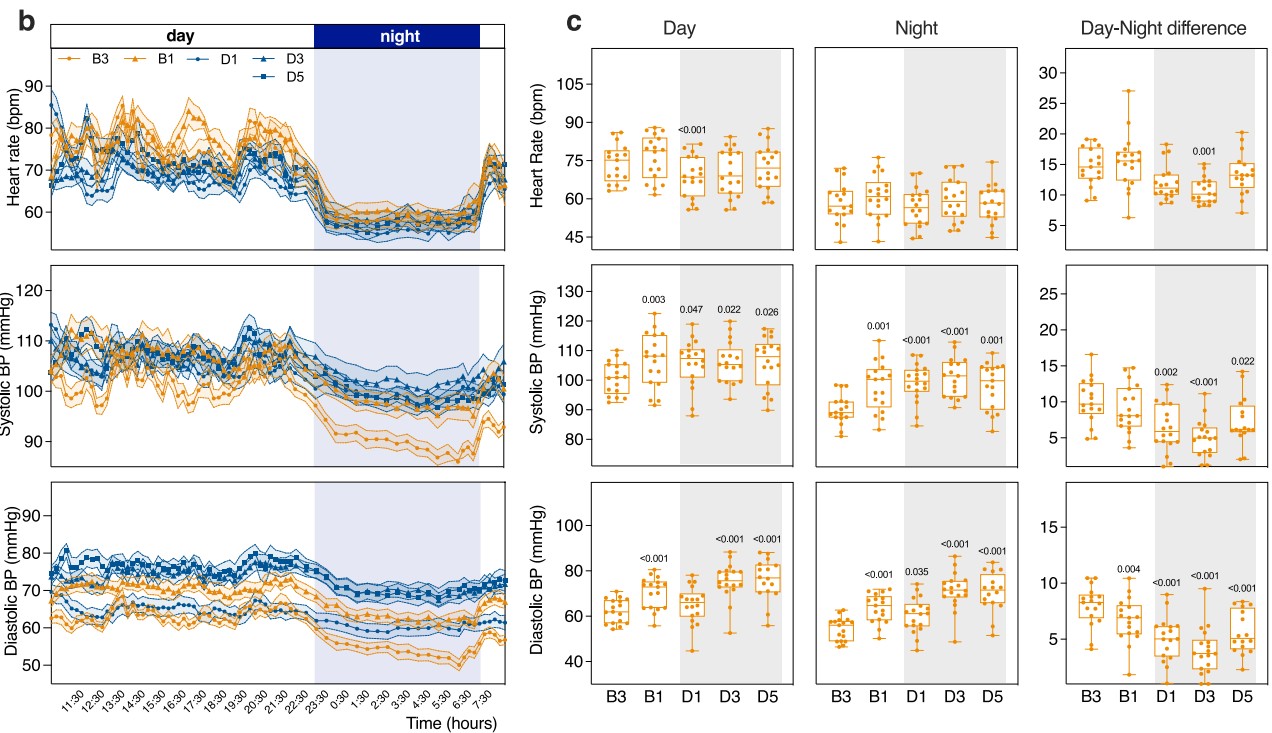

### Core body temperature continuous recording

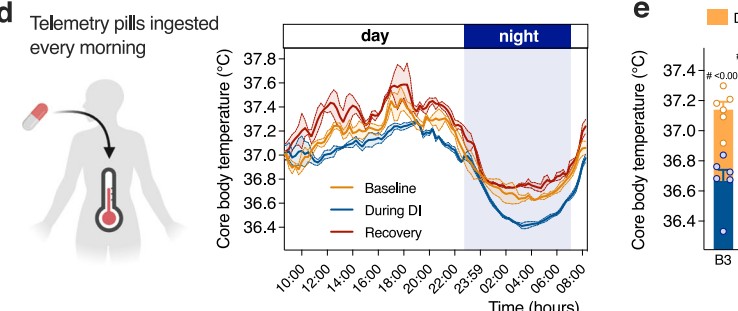

change significantly during DI, and plasma proteins increased significantly after D1 to reach +8 ± 1% at D5 ($P < 0.001$) (Fig. 7c; Supplementary Table 3; Supplementary Table 4). Specifically, plasma albumin increased significantly at D3 (+8 ± 2%, $P = 0.011$) (Fig. 7c). To assess the metabolic stress state, we measured 24 h urinary cortisol, serum thyroxin (free T4) and thyroid stimulating hormone (TSH). Daily urinary cortisol excretion, as well as urinary cortisol normalized by creatinine, did not change during DI, whereas serum T4 and TSH increased (T4: +18 ± 2%, $P < 0.001$; TSH: +46 ± 6%, $P < 0.001$) (Fig. 6b;

Supplementary Tables 3 and 4), suggesting the onset of an overall slight metabolic stress.

Compared to baseline, fasting plasma triglycerides (TG; +42 ± 7%, $P < 0.001$), LDL cholesterol (+15 ± 2%, $P < 0.001$) and non-esterified fatty acid (NEFA) (+48 ± 15%, $P = 0.007$) increased at D5 (Fig. 7a, b; Supplementary Table 4). Fasting plasma HDL also increased at D3 (+6 ± 2%, $P = 0.022$), but then returned to basal values at D5 (Fig. 7a). The atherogenic index of plasma (AIP), highly predictive of cardiovascular risk, increased from −0.36 ± 0.04 at baseline to −0.21 ± 0.03 at

**Fig. 5 | DI drastically decreases activity, dampens circadian variations of heart rate and blood pressure, and lowers core body temperature. a** Accelerometry-derived daily physical activity counts estimated by magnitude of vector during the day and the night from the wrist (left) ($n = 16$), and the ankle (right) ($n = 17$). **b** Continuous recording of heart rate and blood pressure averaged over 15 min. **c** Mean heart rate and blood pressure for day (09:30–23:00) and night (23:00–07:00), and the difference between them (right panel) ($n = 18$). **d** Continuous recording of core body temperature averaged over 15 min. The baseline curve represents the mean of B3 to B1, During DI from D1 to D5, and Recovery from R0 to R1 ($n = 6$). **e** Core body temperature for the day (07:00–23:00) and night (23:00–07:00) (left) and mesor (right) ($n = 6$). Bars represent mean ± s.e.m. Box plots indicate minimum, 25th percentile, median, 75th percentile, and

maximum values. **a, e** Two-way RM ANOVA with post-hoc Bonferroni multiple comparison. Asterisks indicate a significant difference ($P < 0.05$) compared with B1, and [#] indicates a significant difference compared with night. Global ANOVA results: wrist activity immersion effect $P < 0.001$ and day/night effect $P < 0.001$, Ankle activity immersion effect $P < 0.001$ and day/night effect $P < 0.001$. **c** One-way RM ANOVA with post hoc Bonferroni multiple comparison compared with B3. Global ANOVA results: heart rate day $P < 0.001$ and night $P = 0.089$ and day-night difference $P < 0.001$, systolic BP day $P = 0.040$ and night $P < 0.001$ and day-night difference $P < 0.001$, diastolic BP day $P < 0.001$ and night $P < 0.001$ and day-night difference $P < 0.001$. **e** Global ANOVA results: CBT° immersion effect $P = 0.04$ and day/night effect $P < 0.001$, Mesor (24 h) $P = 0.03$. BP blood pressure.

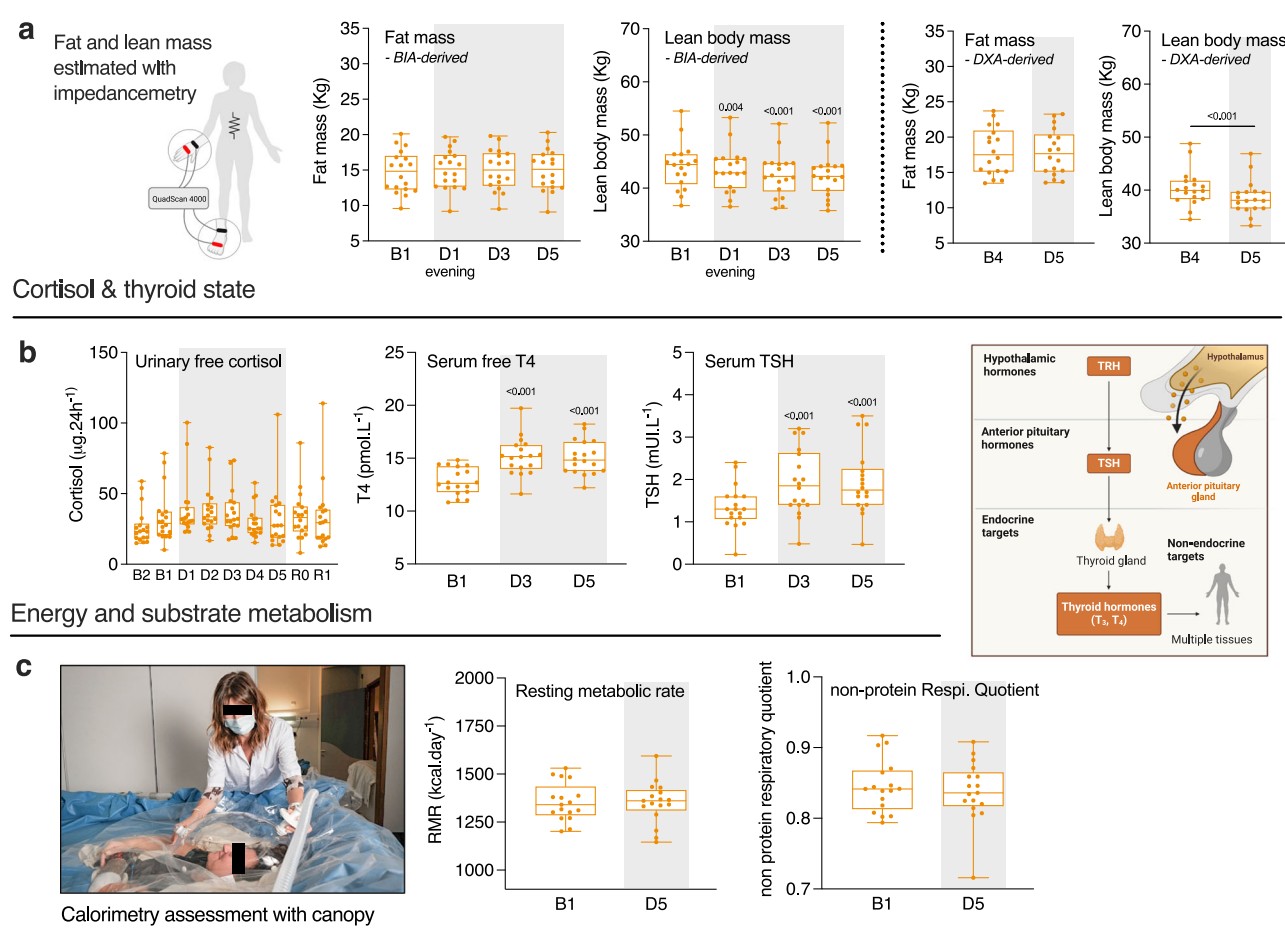

**Fig. 6 | Body composition, cortisol and thyroid state, energy, and substrate metabolism. a** Fat and lean mass estimated via BIA (left), and DXA (right). Measurements were performed supine ($n = 18$). **b** Urinary-free cortisol (left), serum free T4 (middle), and serum TSH (right) ($n = 18$). **c** Resting metabolic rate and respiratory quotient assessed with calorimetry ($n = 17$). Box plots indicate minimum, 25th percentile, median, 75th percentile, and maximum values. One-way RM ANOVA test

with post hoc Bonferroni multiple comparison compared with B1. Global ANOVA results: **a** Fat mass $P = 0.012$, lean body mass $P < 0.001$. **b** Urinary free cortisol $P = 0.237$, serum free T4 $P < 0.001$, serum TSH $P < 0.001$. **a, c** Two-tailed paired Student $t$ test. Shading indicates DI period. BIA bio-impedance analysis, RMR resting metabolic rate.

D5 (Fig. 7b). Fasting plasma adiponectin, an adipocyte-derived secretory protein known for its insulin-sensitizing, anti-inflammatory and anti-atherogenic effect[17,18], significantly decreased at D5 (−15 ± 3%, $P < 0.001$). Fasting glucose was beneath the pre-diabetic threshold (i.e., 6.1 mmol.L⁻¹) in all volunteers at baseline, and was not altered by DI (4.6 ± 0.06 mmol.L⁻¹ at B2 and 4.5 ± 0.07 mmol.L⁻¹ at D3). In response to oral glucose tolerance test (OGTT), two-hour post OGTT blood glucose increased at D3 (6.3 ± 0.35 mmol.L⁻¹) compared to B2

(5.6 ± 0.19 mmol.L⁻¹). Of note, 2 h blood glucose at D3 reached pre-diabetic level (i.e., >7.8 mmol L⁻¹) in three study participants, indicating that some volunteers are more prone to altered glucose tolerance than others (Fig. 7d). Total area under the curve (AUC) for both plasma insulin (+71 ± 14%, from 5515 ± 551 to 9143 ± 1052 µIU.L⁻¹.120 min, $P < 0.001$) and glucose (+18 ± 4%, from 722 ± 29 to 850 ± 39 mmol.L⁻¹.120 min, $P < 0.001$) increased at D3 compared to baseline (Fig. 7d). DI further increased HOMA-IR (homeostasis model

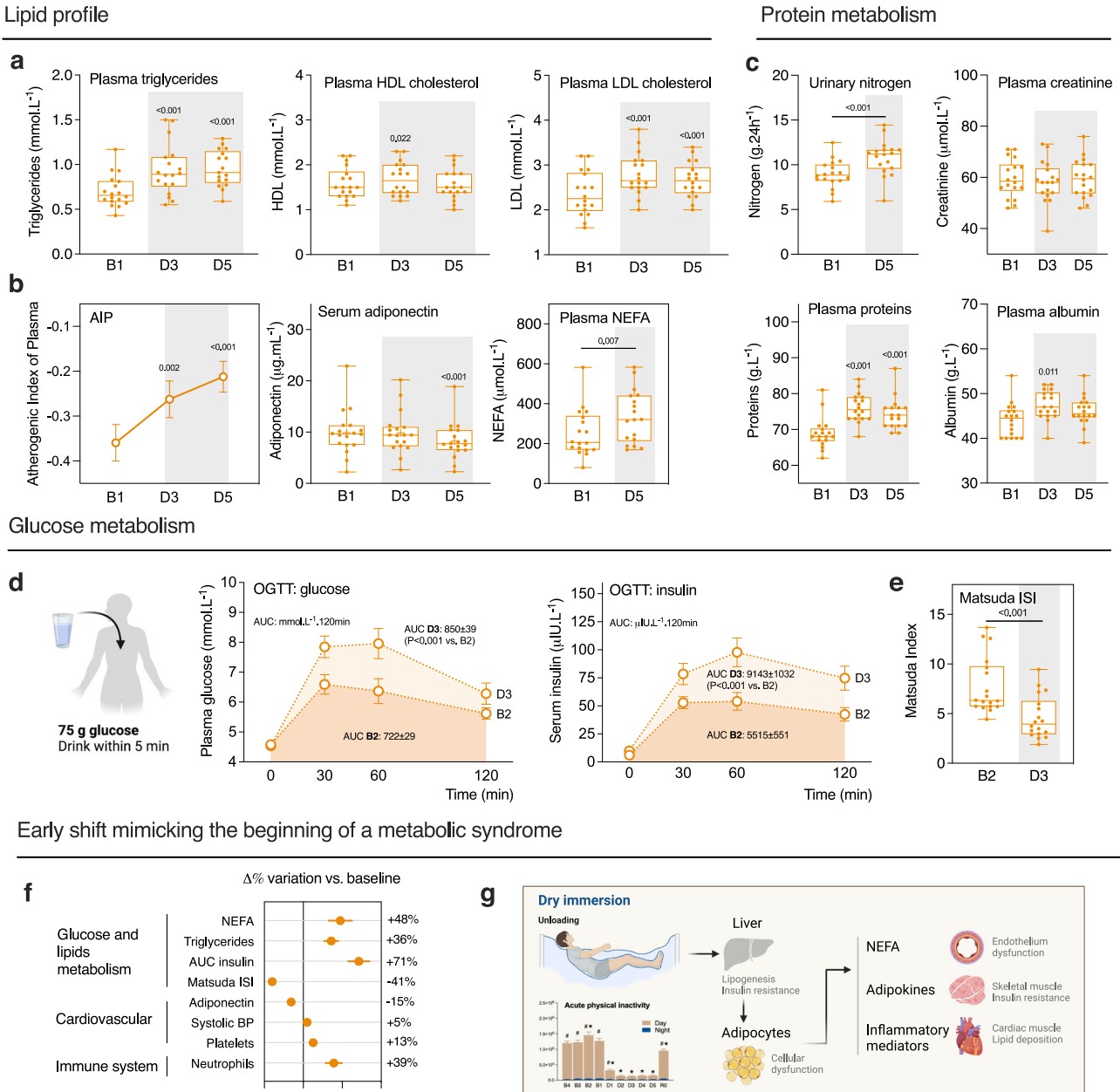

**Fig. 7 | DI induces a sedentary-like metabolism shift. a** Plasma triglycerides (TG; left), HDL (middle) and LDL (right) cholesterol ($n = 18$). **b** Atherogenic index of plasma calculated as log(TG/HDL) (left), plasma adiponectin (middle), and NEFA (right) ($n = 18$). **c** Urinary nitrogen, and plasma creatinine, proteins, and albumin ($n = 18$). **d** Oral glucose tolerance test, performed with 75 g glucose load. Plasma glucose and serum insulin levels at baseline (fasting) and 30, 60, and 120 minutes after glucose load. The total area under the curve (AUC) for glucose and insulin was calculated using the trapezoidal rule ($n = 18$). **e** Matsuda insulin sensitivity index (ISI), calculated as 10.000/√ [fasting glucose × 18 × fasting insulin] × [mean glucose × 18 × mean insulin during OGTT] ($n = 18$). **f** Variations in Δ% of the main parameters involved in the metabolic syndrome at D3 (D5 for NEFA and adiponectin) compared with baseline. **g** Hypothesis schema of the DI effects on metabolism. Bars and lines represent mean ± s.e.m. Box plots indicate minimum, 25th percentile, median, 75th percentile, and maximum. **a**–**d** One-way RM ANOVA compared with B1. Global ANOVA results: **a** Plasma triglycerides $P < 0.001$, HDL $P < 0.001$, LDL $P < 0.001$. **b** AIP $P < 0.001$, adiponectin $P < 0.001$. **c** Creatinine $P = 0.295$, plasma proteins $P < 0.001$, plasma albumin $P = 0.005$. **a**, **c**–**e** Two-tailed paired Student $t$ test for plasma NEFA, urinary nitrogen, OGTT, and Matsuda ISI. Shading indicates DI period. AIP atherogenic index of plasma, AUC area under curve, HDL high-density lipoprotein, ISI insulin sensitivity index, LDL low-density lipoprotein, NEFA non-esterified fatty acids, OGTT oral glucose tolerance test.

assessment of insulin resistance) from 1.56 ± 0.13 at B1 to 2.20 ± 0.19 at D3 (+48 ± 10%, $P = 0.012$) and decreased Matsuda index from 7.8 ± 0.7 to 4.6 ± 0.5 (−41 ± 4%, $P < 0.001$) (Fig. 7e). Altogether these results suggest that DI induced the development of dyslipidemia and reduced insulin sensitivity and glucose tolerance. Here we demonstrated that the evolution of involved parameters indicates a metabolic impairment (Fig. 7f).

## Endothelium and microcirculation state

We assessed soluble endothelial factors. Vascular endothelial growth factor (VEGF) and VEGF-receptor1 (VEGF-R1) levels increased at D3 and D5 compared to baseline (+17 ± 5% and +11 ± 6%, $P = 0.040$, and +34 ± 6% and +15 ± 5%, $P < 0.001$, respectively). The glycoprotein E-selectin, expressed on endothelial cells after activation, varied from 28 ± 2 ng.ml⁻¹ at B1 to 30 ± 10 ng.ml⁻¹ at D5 (Fig. 8a). Average number of

### Endothelial state biomarkers

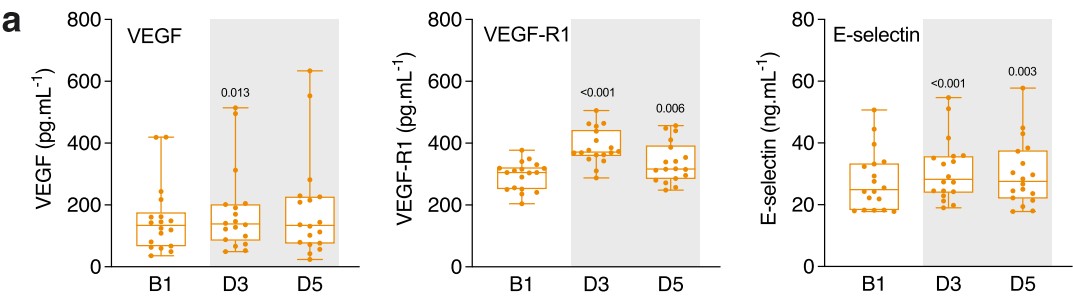

### Nailfold capillaries

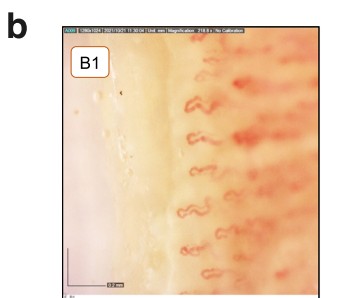 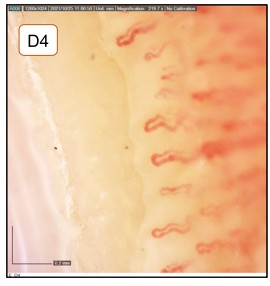 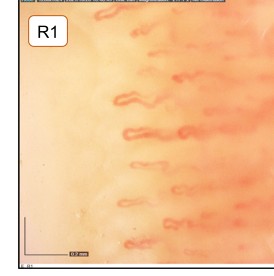 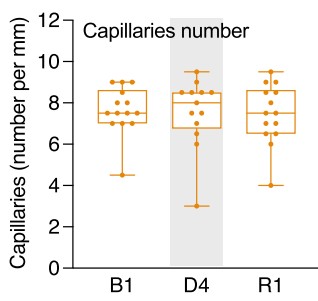

**Fig. 8 | Endothelial and capillary state. a** Soluble VEGF, VEGF-R1 and E-selectin ($n = 18$). **b** Example of capillaroscopy (×200 magnification), and capillary density per mm at B1, D4, and R1 ($n = 15$). Box plots indicate minimum, 25th percentile, median, 75th percentile, and maximum. **a, b** One-way RM ANOVA with post-hoc

Bonferroni multiple comparison compared with B1. Global ANOVA results: **a** VEGF $P = 0.040$, VEGF-R1 $P < 0.001$, E-selectin $P < 0.001$. **b** Capillaries number $P = 0.682$. Shading indicates DI period.

nail fold capillaries per 1 mm length did not change significantly during DI (from $7.6 \pm 0.3$ at B1 to $7.5 \pm 0.4$ at R1) (Fig. 8b).

### Early bone markers

For bone assessment, we were first interested in phosphocalcic metabolism. Total blood calcium significantly increased as early as after the first day of immersion and reached a peak at D3 of $+4.8 \pm 0.7\%$ above baseline. These levels remained elevated during all DI phases then decreased below baseline (BDC) level at R2. Phosphatemia fell to values below the baseline level by $-5-6\%$ at both R0 (morning, still immersed) and R2. Despite these variations, these two parameters remained within the range of physiologically normal concentrations observed for women (Supplementary Table 5). A DI-induced increase in calcemia was accompanied by a time-delayed decrease in intact parathyroid hormone (PTH) levels as shown by $15 \pm 5\%$ lower values at R0 morning when compared to baseline, and tended to remain low until 48 hours of recovery (Supplementary Table 5). Vitamin D was not affected. For bone activity and associated metabolic markers, DI induced a trend towards higher C-terminal crosslink telopeptide type I collagen (CTx) concentrations at R0 morning ($+17 \pm 4\%$, $P = 0.057$) suggesting an increase in bone resorption. Concurrently, procollagen type I N-terminal propeptide (P1NP) significantly decreased progressively during all immersion period ($18 \pm 3\%$ lower values at R0 morning) and remained low even after 48-hours of recovery as a result of decreased formation activity (Fig. 9a, Supplementary Table 5). Similarly, total intact osteocalcin (OC) level decreased as soon as the first 24 hours of immersion and the concentrations at R0 morning were recorded $9 \pm 3\%$ lower than baseline values. During the 48 hours of recovery, they continued to decline until becoming $14 \pm 2\%$ lower than BDC values at R2. Carboxylated osteocalcin (Gla-OC) evolved in a similar way to total intact OC. For its undercarboxylated forms (Glu-OC), the concentration fell to values below the baseline level by

$20 \pm 3\%$ at R2 (Fig. 9a, Supplementary Table 5). Collectively, these data suggest that the DI induces a decreased bone formation activity coupled with a trend for increase in bone resorption (as represented in Fig. 9b). Of note, total bone mineral density was not significantly modified ($1.166 \pm 0.020$ g.cm$^{-2}$ at B4 vs. $1.158 \pm 0.019$ g.cm$^{-2}$ at D5, $P = 0.102$).

### DI effects and menstrual cycle

Cycle phase was deduced from blood progesterone and estradiol dynamics. Individual cycle phases respective to protocol are indicated in Supplementary Fig. 3. Our data do not show a significant impact of the menstrual cycle on the changes induced by DI. Participants with maximal blood progesterone during 5-day DI > 2 μg.L$^{-1}$, set as follicular-to-luteal phase threshold ($n = 6$), were not different in responses to DI. Similarly, those with maximal blood estrogen during 5-day DI ≥ 200 ng.L$^{-1}$ ($n = 7$) did not differ.

## Discussion

Dry immersion, a whole-body thermoneutral immersion with a waterproof film separating the subject from the water, creates a unique supportless environment characterized by buoyancy-induced gravitational unloading, constant hydrostatic compression of superficial tissues with swift fluid centralization and headward fluid shift, constant ambient temperature, constant body position, and constant physical inactivity with acute decrease in postural muscle load and body movements. These factors inherent in the immersion are mainly responsible for the multi-system deconditioning occurring during this microgravity simulation. This comprehensive ESA Vivaldi study investigates multi-system physiological changes in women during a 5-day microgravity simulation through DI technique.

Concerning the main findings, 5-day strict DI in healthy women of 22–39 yrs was well tolerated. We observed a rapid resetting of water-

## Bone metabolism state

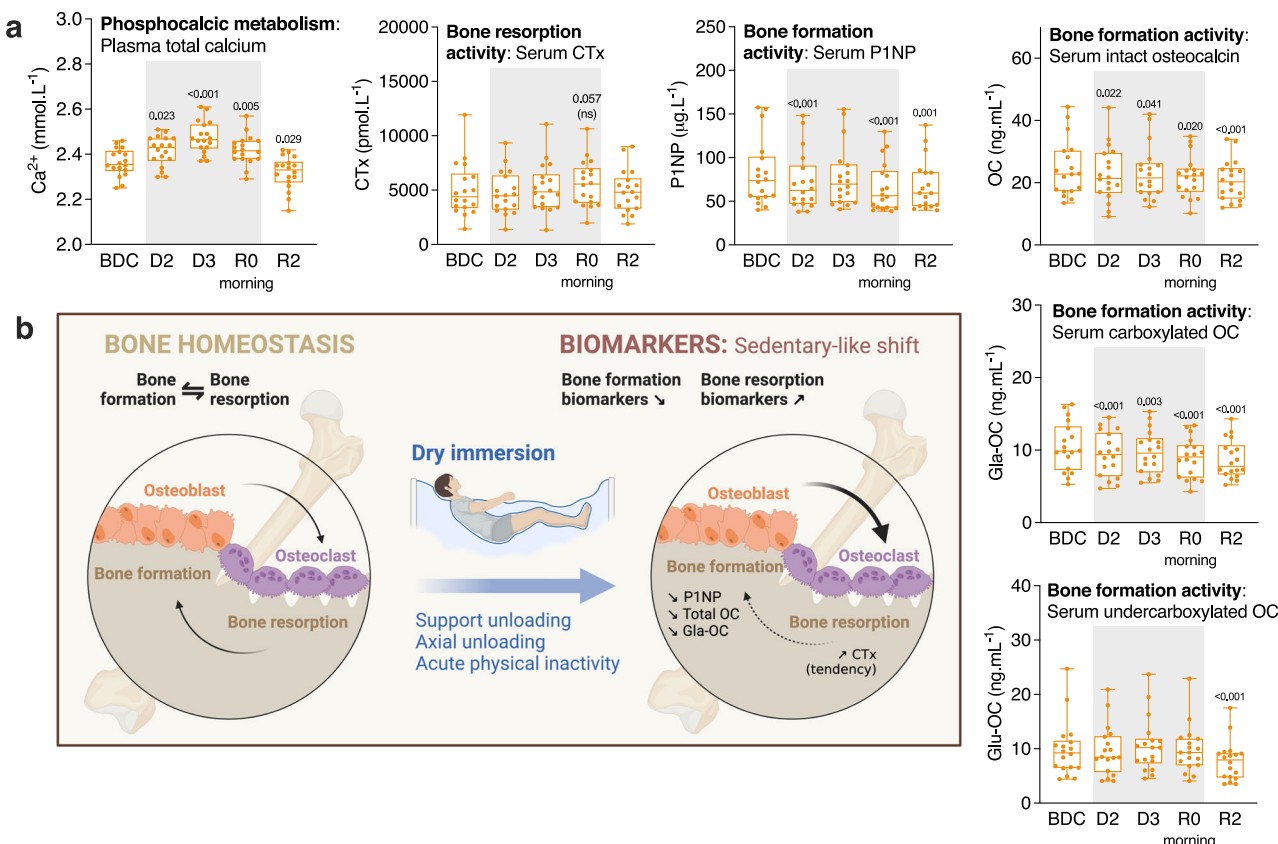

**Fig. 9 | DI induces a sedentary-like bone metabolism shift. a** Evolution of bone blood markers: total calcium, C-terminal crosslinked telopeptide of type I collagen (CTx), procollagen type I N-terminal propeptide (P1NP), intact osteocalcin (OC), and its carboxylated (Gla-OC) and undercarboxylated (Glu-OC) forms ($n = 18$). **b** Hypothesis schema of the DI effects on bone metabolism loop. Box plots indicate minimum, 25th percentile, median, 75th percentile, and maximum. **a** One-way RM ANOVA with post hoc Bonferroni multiple comparisons compared with BDC. Global ANOVA results: total calcium $P < 0.001$, serum CTx $P < 0.001$, serum P1NP $P < 0.001$, serum intact OC $P < 0.001$, serum Gla-OC $P < 0.001$, serum Glu-OC $P < 0.001$. Shading indicates DI period.

electrolyte homeostasis with a decrease in body weight of about 1.5–2 kg, a reduced overall fluid content (TBW) of 4%, and a hypovolemia of 18–24%. An enlargement in ONSD, used as an indirect marker of intracranial pressure (ICP), was 10–15%. DI induced a marked cardiovascular deconditioning with a decreased OT in response to LBNP steps (41% decrease in OT-index), and an increased heart rate (+12 bpm at first-step ergometry load) and a decreased $\dot{V}O_{2max}$ (−7%) and maximum load (−7%) during aerobic capacity test. DI rapidly induced a sedentarily-like metabolism shifts with a pronounced impairment in glucose tolerance (48% increase in HOMA-IR, 41% decrease in Matsuda ISI, and 71% increase in net insulin response to OGTT at D3) and lipid profile (42% increase in TG, 48% increase in NEFA, and increase in AIP at D5). DI induced a muscle impairment with a diminution in whole-body lean mass of 4–5%, predominant for lower limbs with a diminution in knee extensors muscle force of 8%, and calf muscle changes with 6% muscle volume diminution due to reduced fluid content and maybe beginning atrophy. Serum markers of bone remodeling showed a decreased bone formation (18% decrease in P1NP at R0) coupled with an increased bone resorption (17% increase in CTx at R0). We observed a marked impairment of postural balance control following DI (45% increase in CoP velocity with eyes closed), not totally recovered on R1. DI dampened the circadian variations of blood pressure with a decrease in night dipping from 10 to 5% for SBP, from 13 to 6% for DBP, and from 21 to 15% for HR, and lowered 24-h core temperature of 0.15 °C. Overview of major results is given in Fig. 10.

This strict 5-day immersion was well tolerated, as reported for DI studies with men participants[8,19]. The extent of all reported symptoms was consistent with data published on men[12]. All 18 participants completed the study, no substantial complaints related to fluid shift have been reported, and urinary cortisol level remained stable. Out-of-bath time was also typical for DI experiments[12]. Back pain, associated with general discomfort and sleep decline, has been predictably reported in most participants (13 out of 18). In immersion the heavier pelvis sinks, while the thorax, filled with air, is pushed to the surface, flattening the lumbar curvature and lengthening the spine. Stretching and tension might create a lumbar painful tension point[20]. Altogether, the back pain associated with acute inactivity and relative postural discomfort contributes to sleep reduction. The first night is the most challenging, and then the symptoms gradually diminish. Early back pain and sleep reduction during DI mimic the global discomfort observed in space[21].

In our study, the fluid shift results in the hormonal regulation to normalize the acute expansion of central volume detected by the cardiac stretch receptors. Then a hypovolemia state quickly develops, with −12% of PV 8 h after the onset of immersion, stabilized at −18–24% decrease no later than 48 h after the onset). During a previous 5-day strict DI in men, similar rapid PV depletion was observed[12] (−9% of PV after the first 8 h, and −15–25% at steady state), contributing to the decrease in orthostatic and exercise tolerance. During spaceflight, an early loss of 500 mL of PV, an initial increase in hemoglobin concentration, and a natriuretic peptide resetting are observed as a result

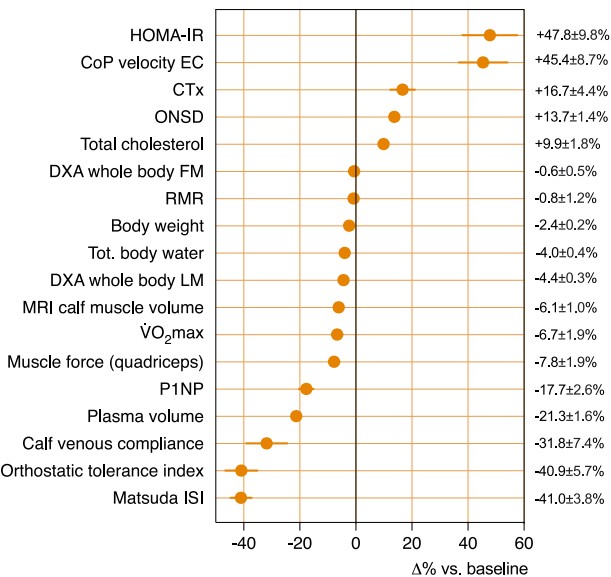

**Fig. 10 | Integrated overview of the multi-systems deconditioning.** Global multi-systems deconditioning for the main tests, represented in Δ% compared with baseline. Results are mean ± s.e.m. HOMA-IR: homeostasis model assessment of insulin resistance ($n = 18$). CoP velocity EC: Center of Pressure velocity during the postural test with eyes closed ($n = 18$). CTx: C-terminal crosslink telopeptide type I collagen ($n = 18$). ONSD optic nerve sheath diameter ($n = 17$). Total cholesterol ($n = 17$). RMR resting metabolic rate ($n = 17$). DXA whole-body FM and LM: Whole-body fat mass and lean mass assessed by dual-energy x-ray absorptiometry ($n = 18$). Tot. body water assessed by bio-impedance ($n = 18$). MRI magnetic resonance imaging. Calf muscle CSA ($n = 12$). $\dot{V}O_{2max}$ maximal oxygen uptake ($n = 18$). Muscle force ($n = 17$). P1NP: Procollagen type I N-terminal propeptide ($n = 18$). Plasma volume evolution ($n = 18$). Calf compliance ($n = 16$). Orthostatic tolerance index ($n = 17$). Matsuda ISI: insulin sensitivity index of Matsuda ($n = 18$).

of the fluid shift[22,23]. The loss of PV and muscular unloading experienced in DI participate to the cardiovascular deconditioning characterized by the triad: orthostatic intolerance−$\dot{V}O_{2max}$ loss−increased heart rate.

To assess ocular changes, we studied the changes in ONSD as an indirect marker of changes in ICP and as a marker of the neuro-ophthalmological changes reported in spaceflights. In this study, we observed an early increase in the ONSD in women (+7% 4 h after the onset of DI), persisting over immersion (+10% at D3 and +14% at D5). Several mechanisms may explain the enlargement in the ONSD. With increase in ICP induced by the thoraco-cephalic fluid shift, the hydrostatic transmittance of cerebrospinal fluid traveling within the subarachnoid space may expand the retrobulbar part, leading to a local ONSD enlargement[24]. Compartmentalization of cerebrospinal fluid within the optic nerve sheath also is one of the considered mechanism of SANS. Venous and lymphatic drainage impairments have also been proposed as an underlying mechanism in ONSD distension. Indeed, a drainage defect may lead to a subarachnoid space distension due to an alteration in cerebrospinal fluid absorption within the orbit. Consecutive long-term ISS missions may also have an impact on the dilation of ONSD, as documented by MRI in a 57-year-old astronaut[25]. In-flight ONSD values from a preliminary study, obtained with 2D ultrasound in 13 astronauts, showed a rise of +11% (+0.9 mm) compared to pre-flight values, with recovery post-flight[26]. A case report has also described an increase in ONSD persisting after the 6-month mission in a 45-year-old male astronaut[27]. These results are consistent with previous studies in healthy men, where we found an elevation in ONSD of +20% during DI[28,29]. Different ground-based studies found significant increase associated with the cephalad fluid shift induced by head-down tilt or DI[30]. Marshall−Goebel et al.[31] showed that LBNP was

able to limit head-down tilt-induced increase in ONSD, illustrating the rapid changes in ONSD with the fluid shift. Noteworthy, some authors found that ONSD was smaller in females, and thus the threshold to define high ICP may be different in females[32]. Several studies tried to define an optimal cutoff value of ONSD in order to predict an intracranial hypertension; but this threshold value is very variable according to the different studies[33–35]. In all, our findings seem to indicate that ONSD values did not reach the threshold observed in those who developed intracranial hypertension. It is likely that fluid shift plays a significant role in the increase in ONSD, taken into account rapid changes and the link with the rapidity of decrease in PV.

The cardiovascular deconditioning includes orthostatic intolerance and $\dot{V}O_{2max}$ loss as key components. Women are more susceptible to orthostatic intolerance after spaceflight. In fact, women astronauts demonstrate a greater loss of PV[36,37], a greater increase in renin during a 6-month ISS mission[38], a greater decrease in baroreflex control of heart rate[36,39] and an hypoadrenergic responsiveness to orthostatic stress[36]. Several reports also note that, on Earth, women predominantly respond to cardiovascular stress with increased heart rate, while men have greater increase in vascular resistance[5,40–42]. OT dropped in our subjects after DI (−41 ± 6% decrease for OT-index). This is consistent with the data of the 5-day DI study in men ($n = 9$ controls) using the same 6 steps LBNP-alone protocol (decrease in orthostatic tolerance time (OTT) from 17.4 ± 1.4 min at baseline to 13.8 ± 4.1 min at R0, with decrease in OT-index of 31 ± 9%)[12], and with a 60-day HDBR in women using pre-syncopal 10-min 80° tilt+LBNP test[43] (−50% for OTT, $n = 8$ controls). After long-duration spaceflights (>4 months), nearly 80% of astronauts suffer from orthostatic intolerance, and after short-term spaceflights, 20% experience presyncope[44]. During long-term bedrest (>35 days) in men, a high range of participants is considered intolerant to an orthostatic stress by tilt or stand test, from 25% to nearly 60%[45–49]. The sample remains however quite small in most of these studies ($n < 10$ participants in control groups). In the present study, 5 days of simulated microgravity led to a decrease in aerobic performance observed by a 7% loss of $\dot{V}O_{2max}$. Participants showed an 11% increase in first-step heart rate during aerobic test, this could be interpreted as a part of the cardiovascular deconditioning. This is in line with results from spaceflights and bedrest studies, as a multi-system deconditioning appears with cardiac atrophy, hypovolemia, and muscle metabolic changes, all contributing to the reduction in exercise performance[49–51]. As a result, microgravity exposure induces an 'ageing-like' cardiovascular deconditioning with loss of physical fitness[52].

Concerning sensorimotor deconditioning, our results demonstrate that 5 days of strict DI decrease postural stability and balance control. Review papers by Navasiolava et al. and Tomilovskaya et al. summarize the findings on sensorimotor system associated with DI[8,19]. DI has a profound impact on sensorimotor control, with decreased accuracy of movement control, altered cortical organization of voluntary movements, and impaired systems of posture and locomotion control. The primary contributors to these changes are support unloading, axial unloading, and muscular unloading. Reduced afferentation leads to functional denervation, with hypersensitivity to proprioceptive and vestibular signals. Postural stability post-DI is similar to that observed post-flight, especially when the task is complicated by unstable support (foam) and the absence of visual feedback (eyes closed). Viguier et al. found ~+45% increase of path length in static position with eyes open after 60 days in HDBR in women[53]. To counteract this deconditioning, it has been shown that treadmill exercise within LBNP significantly attenuated losses of standing rail balance time by 63% in men after 30 days HDBR[54].

The lower limbs suffer from disuse during spaceflight since astronauts in space mostly use their arms to navigate onboard and perform daily tasks. Thus, a musculoskeletal deconditioning appears more pronounced in the lower limbs, particularly in postural muscles

such as the soleus. In this study, muscle disuse during DI participates in the early loss of calf muscle volume and quadriceps strength. However, similarity in volume loss for muscle and skin (5–6% as measured by MRI) indicates the contribution of dehydration process. During a previous 3-day DI in men ($n = 12$), a decrease of −9% in quadriceps maximal voluntary contraction has been observed, concomitant with a significant atrophy of type I muscle fibers and an increase in neural cellular adhesion molecule (NCAM+) muscle fibers, suggesting an early denervation process[55]. During two weeks of leg immobilization, women ($n = 14$) showed a greater decrease in specific strength compared with men ($n = 13$), but not in whole muscle atrophy[56]. This is consistent with what happens during spaceflight and bedrest, where muscle mass, strength, and endurance are decreased[57–59]. In space, the lower limb muscular deconditioning coupled with the hypovolemia lead to an increase in venous compliance, and contribute to the orthostatic intolerance experienced by astronauts when returning to Earth[60]. In our study, we observed a reversible decrease in calf compliance during DI. It could be due to the squeezing effect of the water mass around the body, acting like compression therapy for venous insufficiency.

We recorded circadian rhythms of 24 h BP, which is slightly increased in our study, contrary to 8–10 mmHg decrease observed in long-duration spaceflight[61]. Interestingly, this increase began the day before the actual immersion, maybe suggesting pre-launch stress. Day-night rhythms of HR and BP were visibly preserved, however, the night dip in HR and BP was slightly flattened. Flattening of day-night oscillations for HR is in agreement with findings on 5-, 21-, and 60-day HDBR, revealing such flattening progressing with the duration of bedrest[62]. In line with the early lower limb musculoskeletal deconditioning mentioned above, continuous 24 h recording of physical activity and HR demonstrates a sedentarily-like lifestyle especially pronounced on the legs. The acute decrease in activity counts, slight decrease in 24-h HR and its day-night variations closely mimics the marked reduction in ankle activity and 24-h HR during the first 3 weeks of spaceflight[63]. During this DI we reported a decrease in core temperature, presumably due to extreme inactivity and increased convective heat loss through the surrounding water. In space, convective and evaporative heat losses are impaired due to the absence of gravity. Thermoregulation can thus become critical with a risk of hyperthermia, especially during exercise. Moreover, the homeostatic set-point of core temperature itself might be modified in mission, and some astronauts report thermal discomfort. Using core temperature estimation by a heat flux sensor positioned on the forehead[15], impaired thermoregulation was reported in astronauts ($n = 11$) at rest (increase in core temperature of +1 °C) and during exercise (higher and faster rise) in 6-month mission.

Concerning metabolic alterations, the loss of lean body mass is a systematic observation in response to simulated and actual microgravity that is proportional to the duration of the exposure[64]. Five days of dry immersion induced pronounced deconditioning in women, as it was sufficient to reduce fat-free mass and trigger greater protein catabolism. It was further associated with the development of metabolic alterations including the onset of metabolic stress, dyslipidemia, and decreases in insulin sensitivity and glucose control. Prior HDBR and inflight studies showed that the organism perceives weightlessness as a stress. During Spacelab Life Sciences-1 (SLS$_1$) and SLS$_2$ missions, the whole protein turnover, i.e., both protein synthesis and degradation[65], acute phase hepatic proteins, cortisol secretion, and biomarkers of systemic inflammation (IL-1 and IL-10) increased during the first day of flight. Medium-term HDBR (42 days) reported increases in urinary excretion of cortisol, urea, creatinine, and growth hormone (GH) to last around 4–5 weeks[66]. Decreases in insulin sensitivity were reported as soon as after 3 days of HDBR[67] and DI in male adults[68]. Indeed, three days of strict DI increased HOMA-IR by $43 \pm 11\%$, net insulin response by $72 \pm 23\%$, and decreased Matsuda ISI by $31 \pm 7\%$[68]. This reduced

insulin sensitivity is likely associated with a blunted insulin-stimulated glucose disposal as shown recently after 3 days of HDBR in healthy males[69]. We observed no shift in the use at fast of substrate from lipid toward carbohydrate oxidation after 5 days of DI. Alterations in substrate metabolism may take longer to be apparent. Indeed, while 3 days of HDBR failed to detect changes in insulin-stimulated carbohydrate and fat oxidation, changes in fuel selection were reported after 7 days of HDBR[70] and after the long-term HDBR[71,72] in both male and female adults. In agreement with results obtained in long-term HDBR[72], pronounced increases in fasting plasma TG, LDL, NEFA, and AIP were finally observed suggesting atherogenic effects of 5 days of DI. This raises a question of cardiovascular risk for long-term missions. Of note, while sharp decreases in plasma progesterone concentration were reported after 5 days of DI compared to ambulatory conditions on the same day of the cycle in healthy females[73], menstrual cycle was not controlled in our female participants. Although this limitation needs to be acknowledged, we did not observe differential metabolic responses to DI between individuals with high versus low plasma progesterone and estradiol levels, suggesting that the effects of DI may be stronger than hormonal modifications. In addition, a drop in progesterone would be expected to counteract the metabolic effects of DI[74].

In our study, a probable alteration of endothelial survival/apoptosis equilibrium is suggested by the observed 12% increase in sVEGF along with 15% increase in VEGF-R1. VEGF transfers survival and proliferation signals towards the endothelial cells, thus characterizing their anti-apoptotic tone, whereas VEGF-R1 sequesters VEGF from signaling receptors and forms non-signaling complexes with VEGF-R2, behaving as natural endogenous inhibitor of VEGF. Soluble E-selectin was not substantially modified, suggesting absence of important endothelial inflammatory activation. These results are consistent with data from 7-day DI in men[75].

The skeletal unloading negatively impacts astronauts' musculoskeletal system when exposed to microgravity. Serum levels of CTx have been shown to increase as early as the 8th day of spaceflight[74] followed by a drop in bone formation markers (P1NP, OC) after the first week of spaceflight[73–76]. The decrease in bone mineral density is particularly pronounced at the femoral neck and proximal femur (around −8%) after 4.5–7 months of spaceflight, as shown by DXA[73]. Bone mineral density and bone-associated biomarkers did not differ between male and female astronauts[77]. Here, we demonstrate in females a decrease in bone formation markers (P1NP, OC, Gla-OC) and a trend towards an increase in bone resorption (CTx) after 48 h of DI, indicating an imbalance with rapid bone adaptation due to acute unloading. These results confirm that short-term DI as a model of microgravity challenges bone remodeling activity in females. In men, in 5-day DI ($n = 9$ men controls), variations in total Ca (+5%), PTH (−18%), and bone formation markers (−15% for P1NP and −15% for intact OC) were similar[76]. As for CTx (resorption marker), it was 13% increased in 3-day DI (12 men)[77], but unmodified in 5-day DI[76], compared to 17% increase trend in women. Finally, the increased calcium excretion secondary to bone loss during spaceflight is associated with an increased risk of renal oxalate stone formation in men and women, without sex differences[77]. In this study, we reported an increased blood calcium and a decreased water balance, mimicking changes observed during spaceflight and further emphasizing an environment favoring renal stone development.

A methodological comparison can be done with a first DI study in women of reproductive age, Naiad-2020, which was conducted in Russia in September-November 2020[9–11]. This pilot DI lasted for 3 days and included 6 healthy women of 24–39 yrs. Special home-made universal portable device for mechanical urine collection in women was designed (for details see Tomilovskaya et al.[9]), allowing for most cases urination in bath (with a pillow under the back and shoulders), though sometimes a solid support was needed. During immersion, subjects spent 20–30 min.day$^{-1}$ out-of-bath, including 7–9 min.day$^{-1}$ sitting and

standing (non-strict DI protocol). Regarding to menstrual cycle, the onset of Naiad DI was standardized—all 6 participants began DI at the 7th day of the cycle and completed it at the 10th day. Water intake was ad libitum. Diet was standardized from B1 to R0 with caloric intake of 2200–2300 kcal.day$^{-1}$; taste preferences of the participants were taken into account, and some adaptations from the proposed menu were allowed to maintain a comfortable psychophysiological state during DI.

Concerning limitations, to our knowledge 18 participants is the largest sample size ever used for a single DI protocol without countermeasures, though it remains limited. Even if random error for some measurements cannot be excluded, in this study we were able to identify significant changes for major functions with underlying physiological explication or hypotheses. Individual data points are shown to give the reader an exact idea on inter-individual variability. Various confounders could potentially impact results under DI, such as individual predisposition to stress and boredom, despite high motivation of all the subjects to do their best. Loneliness could also appear, especially for two subjects without roommates who underwent DI alone. The staff was particularly careful to these subjects and spent more time with them, being more available with twice less workload. They were not different from the rest of the group in discomfort, sleep quality, and 24-h cortisol. Another limitation of our study is that the onset of DI was not synchronized with menstrual cycle phase. Individual cycle phases respective to protocol were deduced from sex hormones profiles. As already stated, our data did not show a significant impact of the menstrual cycle on DI effects, but this impact remains possible. Furthermore, the question of the effect of DI on the cycle and more generally on reproductive system was not explored in our study. Interestingly, the very recent data on the effect of 5-day dry immersion on the female reproductive system[73] show a sharp decrease in progesterone after immersion.

To improve comparability of obtained data with future studies, we might consider recruitment of participants within similar ranges of age, fitness ($\dot{V}O_{2max}$), anthropometry (height, BMI), naive to DI, and preferably also with no previous participation to any spaceflight analogs. To limit potential bias, it would be also useful for future studies to better characterize the subjects before experiment, not only via questionnaires/interview as done in the present study, but also via measurement of their usual daily lifestyle, i.e., physical activity level by actigraphy, food habits by dietary survey.

In conclusion, obtained multi-system and multi-dimensional data can now serve as a reference for future space physiology research in women using the DI model, and for developing countermeasures in the scope of future professional mixed crew onboard new space stations (e.g., NASA-ESA-CSA-JAXA Gateway station) and lunar base, as well as private commercial flights with paying customers[24]. Similar to the NASA Twins Study[78], our study should be considered framework-defining. It is the first to provide a physiological and biological dataset and an integrative multi-system assessment of strict DI effects in women. We demonstrated the ability of DI model to closely mimic human exposure to a microgravity environment. Compared to known data from dry immersion studies with male participants, only slight sex differences are revealed in response to DI, such as a smaller increase in ONSD (10–15% vs 20% in men—data from a 3-day DI[29] and from a 5-day DI[28], more pronounced orthostatic intolerance (41% decrease in OT-index vs 31% in men—recalculated from personal data for control group of a 5-day DI[12], to allow comparison for this parameter), and somewhat greater impairment in glucose metabolism (41% decrease in Matsuda index vs 31% in men—recalculated from personal data of a 3-day DI[68], to allow comparison for this parameter). But the overall direction of changes is the same, and the results on women are largely comparable to those on men. In particular, loss in PV, body weight, TBW, lean mass, $\dot{V}O_{2max}$ match perfectly. This suggests that individual differences may be more influential than sex differences in terms of physiological deconditioning caused by microgravity simulation

through DI. Further research on simulated microgravity by DI is needed, particularly focusing on the impact of the individual variances, and must be complemented with sex-based evaluation of countermeasures.

## Methods

### Ethical approval
The experimental protocol conformed to the standards set by the Declaration of Helsinki and was approved by the National Ethic Committee (CPP Ile de France II: 5 July 2021, no. ID RCB: 2021-A00705-36) and French Health Authorities (ANSM: 31 May 2021). ClinicalTrials.gov Identifier: NCT05043974. All participants provided their written informed consent. Participants whose images appear in this article have given their consent to publish images.

### Participants
Power-based calculation of the number of subjects is not directly applicable for such explorative studies, so to determine sample size, we based on combined considerations of physiological assessments, taking into account already known effects of DI. Thus, considering data obtained with previous MEDES DI in men[12,68], minimal sample to expect statistically significant difference between Pre- and Post-DI (with power 80% and alpha level 0.05), could be estimated as $n = 5$ for PV evolution (effect size 2.01), $n = 11$ for OT (effect size 0.96), $n = 15$ for glucose tolerance (effect size 0.8), $n = 19$ for $\dot{V}O_{2max}$ (effect size 0.69). Therefore, a total of 20 subjects was deemed necessary for the study and approved by the National Ethic Committee.

The call for candidates through advertisements via Internet and media started after regulatory approval. Selection consisted of preliminary screening, followed by medical examinations. The inclusion criteria were: age 20–40 years old; body mass index (BMI) between 20 and 26 kg m$^{-2}$; no sedentary nor high-level athletes with $\dot{V}O_{2max}$ between 30 and 55 mL kg$^{-1}$ min$^{-1}$; height between 158 and 180 cm; no combined estroprogestative contraception (progestogen-only pills, intrauterine devices, implants or absence of contraception allowed); certified as healthy by a comprehensive clinical assessment including a detailed medical history and complete physical examination; covered by a Health insurance. The non-inclusion criteria were: any chronic disease; acute infection; cardiovascular, neurological (in particular vestibular disorders and orthostatic hypotension), ear-nose-throat, orthopedic or musculoskeletal disorders; tobacco, alcohol, or drug addiction; medications except for the accepted means of contraception.

Twenty healthy women were recruited with signature of the informed consent (date of first and last participant recruitment: July 27th 2021 and November 26th 2021). One participant left the protocol on the first day of immersion due to technical issue (technical problem with bath lifting platform which could not be fixed quickly without emptying the bath leading to the decision to stop the study for this participant), and another could not be included for regulatory reasons (this participant unexpectedly was still during the exclusion period following participation in a previous unrelated clinical trial). They were excluded from analysis. A total of 18 participants between the ages of 22 and 39 were finally included in the study (mean±s.e.m at baseline, age: $29 \pm 1$ yr; height: $164.8 \pm 1.4$ cm; weight: $59.3 \pm 1.5$ kg; BMI: $21.8 \pm 0.4$ kg.m$^{-2}$; aerobic fitness: $38.7 \pm 1.5$ mL.kg$^{-1}$.min$^{-1}$); resting metabolic rate: $1334 \pm 25$ kcal day$^{-1}$; morning heart rate: $64 \pm 2$ bpm; morning blood pressure: $109 \pm 3/64 \pm 1$ mmHg; baseline participants characteristics are summarized in Supplementary Table 1). According to regulations on healthy volunteers in biomedical research, participants received a financial compensation of 2200 euros.

### General protocol
The study was conducted at the MEDES space clinic, Toulouse, France from 20 Sept. 2021 to 10 Dec. 2021. Participants arrived in the evening

of B5 (B - before) and left in the morning of R2 (R - recovery). The experimental protocol included 4 days of ambulatory baseline measurements before immersion (B4 to B1), 5 days (120 h) of DI (D1 to D5), and 2 days of ambulatory recovery (R0, R1) (Fig. 1d). We were interested in the early phase of adaptation to microgravity (corresponding to the initial phase of flight). Furthermore, we consider that towards D3-D4 of DI the new steady state, i.e., equilibrium is globally established following the acute transition (though for some systems this establishing is ongoing). As a result, a 5-day DI appears to be the best compromise to study this early phase. To minimize deconditioning during the baseline, participants wore pedometer and were asked to walk 8 to 10 thousand steps daily from B4 to B1. Two participants in two separate baths underwent strict DI simultaneously in the same room (except for two persons who were alone because of protocol exclusion of their pairs). The strict DI protocol did not permit participants to rise at all, as the return to a vertical position inevitably induces a Z axis gravitational gradient. This procedure permitted conditions to be closer to what is experienced during spaceflight. However, daily hygiene, weighing and some specific measurements required extraction from the bath (with built-in lifting platform). Total out-of-bath supine time for the 120 h of immersion was mean (SD) 10.4 (1.5) h (Supplementary Fig. 1c). During these brief out-of-bath periods participants maintained the 6° head-down position, except for the lower limb venous test, DXA and MRI procedures during which the participants remained horizontal to match with the pre-DI test position.

Body weight, blood pressure, and heart rate were measured daily. Onset (at D1) and end (at R0) of DI both occurred at ~09:30, therefore morning measurements and samplings at D1 were performed before immersion, and on R0—still under immersion. Water temperature was continuously maintained thermoneutral (32-34 °C) and recorded. Light-off period was set at 23:00–07:00. Water intake was ad libitum in frames of 35–60 mL.kg⁻¹.day⁻¹, and recorded. On each experiment day meals were identical for all participants. Daily caloric intake was adjusted to individual basal resting metabolic rate (BMR) measured at B4 by canopy dilution respirometry (Quark, Cosmed, Italy). Energy intake corresponded to 160% BMR for baseline and recovery (average 2100 kcal), and 130% BMR for the immersion period (average 1700 kcal). Daily intake for sodium and potassium was ~3–4 g. Daily nutrition is detailed in supplementary data (Supplementary Table 6).

This protocol was designed as a Single-group assignment and open-label study (no blinding was imposed by the protocol for data collection and analyses).

## The DI model

DI involves immersing participants in thermoneutral water covered with an elastic waterproof fabric (Fig. 1). Participants, surrounded by fabric, are freely suspended in the water mass, water envelops them from all sides, but they remain dry. The film is thin and sufficiently large, so the hydrostatic pressure is equally distributed across the surface of the body (Fig. 1c). This absence of support gradient simulates supportlessness, an essential condition of weightlessness. Supportlessness and gravitational unloading due to buoyancy are key differences with HDBR, bringing DI closer to microgravity. For a relatively short duration, the model can faithfully reproduce most physiological effects of actual microgravity[8,19].

## Questionnaires

General discomfort and Back pain (morning and evening), and Quality of night sleeping (morning) were assessed using 0-to-10 visual analog scale daily from B4 to R1. Complaints related to fluid shift (morning and evening): headache, heavy head, nasal congestion, eye pain, face swelling sensation, hoarse voice, impaired hearing or vision, and nausea—were assessed using 0-to-5 scale daily from B4 to R1.

## Blood sampling

Antecubital venous blood samples were collected before (in the morning at B4, B2, B1, and D1), during (in the evening at D1, in the morning at D2, D3, D5, and R0), and after immersion (in the morning at R1 and R2). Morning blood sampling was performed before breakfast. Plasma and serum samples were analyzed for blood count, hemostasis markers, blood chemistry, metabolic state markers, vascular markers, bone markers, hormones (volume-regulating, sex, thyroid). The AIP, a marker of lipoprotein particle size, was calculated from TG and high-density lipoprotein cholesterol (HDL) molar concentration as log(TG/HDL). The complete list of assays and methods of assessment is described in Supplementary Table 7.

## Urine sampling

24-hour urine collection started in the morning with the 2nd void and ended after the first void the following day. All samples were weighed and pooled into the storage container kept at +4 °C. For each 24-h pool, aliquots were prepared and frozen at −80 °C. Partial water balance, defined as the difference between consumed water and urine volume, was calculated. Urine samples were analyzed for urinary chemistry (Na⁺, K⁺, Cl⁻, osmolality, creatinine, urea), urinary free cortisol, antidiuretic hormone (ADH), and 24-h urine nitrogen. The complete list of assays and methods of assessment is described in Supplementary Table 7.

## PV evolution

Percent change in PV on D1 evening, D3-morning, D5-morning, R0 morning vs. baseline (D1-morning before the onset of immersion) was estimated using hemoglobin (Hb) (HemoCue Hb-201® system) and hematocrit (Hct) (microcentrifugation) count[79]:

$$\Delta PV(\%) = \frac{[HbB] \times (1 - 0.01[Hcti])}{[Hbi] \times (1 - 0.01[HctB])} \times 100 - 100 \tag{1}$$

Where HbB and HctB are baseline Hb and Hct levels, and Hbi and Hcti are Hb and Hct on further days (D1 evening, D3, D5, and R0).

Percent change in PV calculated indirectly using plasma protein count was estimated as follows:

$$\Delta PV(\%) = \frac{[Prot\ B]}{[Prot\ i]} \times 100 - 100 \tag{2}$$

Where Prot B is protein concentration at baseline, and Prot i are the corresponding values on D1 evening, D3, D5.

## ONSD: ultrasonography

Ocular examination was performed at B2, D1, D3, D5, and R1 by investigators trained in ocular ultrasonography to assess the ONSD, an indirect marker of intracranial pressure (ICP). Ultrasound measurements were performed with a linear high-frequency probe (Orcheo-Lite, Sonoscanner, Paris, France) and analyzed using Ondina software (v.1.1.2, Sonoscanner, Paris, France). The probe was placed on the closed eyelid and adjusted to obtain an appropriate visualization of the optic nerve. The assessment was realized in a two-dimensional mode and the ONSD was measured 3 mm behind the right and left ocular globe in the sagittal plane. The final measure is the mean of both right and left ONSD.

## OT: LBNP test

We have chosen the LBNP test to assess cardiovascular deconditioning and tolerance to orthostatic challenges before and after immersion. The LBNP induces fluid shifts and related hemodynamic responses similar to Head-Up Tilt Test (HUTT) due to fluid transfer toward the lower part of the body. The advantage is the possibility of applying

gradual pressure levels to give a more sensitive assessment of OT than an all-or-nothing HUTT. As the participant remains lying in the device, the test does not require the contraction from lower limb muscles. This test was conducted in the morning (09:00–10:00) in a temperature-controlled room (range 22–26 °C) on B3 and immediately following the end of DI on R0 (first orthostatic challenge after DI), with independent medical supervision for participant safety. Participants remained supine and baseline data were recorded for 5 min. After that, LBNP was applied with steps of −10 mmHg every 3 min. The test was considered finished after completing the −60 mmHg step (completing the totality of 18 min of LBNP). Test was stopped earlier upon the appearance of pre-syncopal signs (pallor, sweating, feeling faint, nausea, dizziness…), a request to stop, or occurrence of one or more stopping criteria: quick and persistent decrease in SBP of 35 mmHg or more, or SBP ≤ 70 mmHg; abrupt variation in HR of at least 15 bpm in stable period (excluding step change phases), or important tachycardia with an increase of >50 bpm; clinically relevant cardiac rhythm disorders. During the LBNP test, finger blood pressure (Nexfin, BMeye, United States) and standard electrocardiogram (ECG; Biopac, ECG 100 C, United States) were recorded continuously. OT time was assessed, and the cumulative OT index was calculated by summing the product of duration and negative pressure at each completely or partially accomplished gradation of LBNP.

### Aerobic performance: $\dot{V}O_{2max}$ test
Maximal oxygen uptake ($\dot{V}O_{2max}$) was assessed via incremental dynamic leg exercise test on a cycle ergometer (Ergometrics 800 S, Ergoline, Bitz, Germany) in the sitting position, at B2 and R0 in the afternoon (16:00–17:00), with independent medical supervision for participant's safety. ECG and blood pressure were monitored. Breath-by-breath $\dot{V}O_2$ was recorded with an Oxycon Pro metabolic cart (E. Jaeger, Hochberg, Germany) using a face mask. Data were analyzed using LabManager software (v. 5.3.04, Cardinal Health, Germany). Participants cycled at minimum 70 rpm for 3 minutes at 45 W, followed by an increase of 15 W every minute until they could no longer maintain the cadence of 70 rpm (peak exertion reached). The exercise test took ~12–16 minutes to complete. Following this test, the participant pedaled on the cycle ergometer at a low work rate (~45 W) for an adequate recovery from peak exertion. In this paper, we report the results for Maximum load (load during the last step), $\dot{V}O_{2max}$ ($\dot{V}O_2$ average for the last 30 s of maximum load), first-step HR (HR average for the last 30 s of the first-step load) and maximum HR (HR average for the last 30 s of maximum load).

### Postural balance: posturography
Standing balance was assessed at B4, B3, R0, R1 by recording modifications in the position of the CoP on a force platform (Leonardo Mechanograph Ground Reaction Force Plate; Novotec Medical GmbH, Pforzheim, Germany) using Leonardo Mechanography software (v. 4.2, Novotec Medical), as described previously[80]. Tests were conducted at 10:30-11:30.

The conditions were: (1) standing on the foam surface, with eyes open and head fixed; (2) standing on the foam, with eyes closed and head fixed (3) standing on the foam with eyes closed and head movements in pitch by metronome (20 per min). For all three conditions, participants were instructed to stand quietly, with arms comfortably at their sides, feet apart, for up to 30 seconds. There were three trials for each condition in the random order, and the mean values of three trials were calculated. In this paper, we report the results for the mean velocity of CoP, path length of CoP, and standard ellipse area of CoP (including 90% of all CoP points).

### Body mass and composition: bio-impedance and DXA
Bio-impedance measurements by Bodystat QuadScan 4000 (Bodystat Ltd., Isle of Man, United Kingdom) were performed in a supine position

when participants were out-of-bath in the morning at B1 (baseline measurement), in the evening at D1, and morning D3 and D5. Participants were weighed prior to each measurement. To ensure the same electrodes placement for subsequent measurements, their positions were marked on the skin. Whole-body wrist-ankle measurement was used to estimate TBW, ECW, intracellular water (ICW), lean body mass, and fat mass. Segmental impedancemetry was applied to assess volume evolution for arm, leg, calf, and torso segments.

At B4 baseline and D5, DXA assessment of body composition (regional and total lean body mass and fat mass) and bone mineral density was performed by a QDR 4500 W scanner using version software 11.2 (Hologic, Nassy, France).

### Calf muscle and skin CSA: MRI
We performed calf MRI on the 1.5 T MRI (Magnetom Sola, Siemens Healthineers) at B3 and D5 in the evening (20:00–21:00). T1-weighted Dixon technique was used. We performed one pack of 30 axial slices, with slice thickness of 6 mm and inter-slice gap of 1.2 mm. For the CSA measurement, the largest slice of the calf cross-section for both legs was taken into account (the same at B3 and D5 respectively to the distal apex of patella). In this slice, measured volume for muscles and skin was divided by 6 mm to obtain CSA. Twelve participants out of 18 performed MRI. The six other participants were allocated to the sub-study with temperature recording using non-MRI-compatible capsules (see below).

### Muscle force: maximal voluntary contraction
Maximal voluntary isometric contraction (MVC) strength of the knee and ankle flexors and extensors was measured at B4 and R1 using ConTrex® device (Physiomed; Schnaittach, Germany) with Human Kinetics 1.7.5 software. A familiarization session was undertaken before the first measurement.

For all tests, the left leg was tested. The isometric extension and flexion contractions were performed in the sitting position. Participants were firmly strapped to the chair of the ConTrex device, to avoid movement when MVC was tested. MVC was determined at 80° extension of the knee and 0° extension of the ankle. The test protocol was similar for each muscle group; after a short warm-up phase in the neutral position, a series of measurements were recorded with a 30-s recovery interval. Each series consisted of an extension movement followed by an isometric contraction and a flexion movement followed by an isometric contraction. Each contraction was maintained for 5–7 s, and a two-minute recovery period was permitted after every three sets of measurements. To determine the MVC, the maximum force level (Nm) achieved during the test was recorded. Three complete sets of extension/flexion contractions were recorded. If the participant was still improving at the third contraction, successive contractions were recorded until no further improvement was observed.

### Calf venous compliance: venous occlusion plethysmography
Venous function was determined at B1, D4, and R1 in the morning (10:00–11:00) using an Air Plethysmograph APG® 1000 (ACI Corporation, San Marcos, CA, USA). The device consists of a tubular air cuff, positioned around the left calf, inflated to a 6 mmHg pressure by an air pump. During testing, the pressure in the cuff was continuously measured reflecting variations in calf volume. Venous occlusion at 30, 40, and 50 mmHg was performed using a manual pneumatic thigh cuff. For each venous occlusion step, a pressure curve was obtained, with an increase in calf volume followed by a plateau with occlusion, and a decrease in calf volume returning to pre-occlusion values with thigh cuff deflation (as described by Fortrat et al.[60]). Venous occlusion was applied long enough to reach the plateau of the pressure curve as visually estimated by the operator (A.R.), each occlusion step lasted a maximum of 5 min. Venous filling for each occlusion step was

determined as calf volume increase at the plateau, and plotted against occlusion pressure to obtain pressure-to-volume relationships. The slopes of the regression lines of these relationships were used as an indication of venous compliance. Plethysmography data were collected through a Biopac device with AcqKnowledge 5.0 software (Biopac Systems, CA, USA).

## Continuous physical activity: actigraphy

To quantify daily body movements, participants wore accelerometers (Actigraph GT3X+, LLC, Fort Walton Beach, FL) on the non-dominant wrist and ankle from B4 to R1 all the time including sleeping (except for showering). Activity counts at wrist and ankle were sampled over 10-second epochs at three axes (vertical, longitudinal, lateral), and summarized as the Magnitude of vector incorporating counts for all three axes:

$$\text{Magnitude of vector} = \sqrt{(x \text{ counts}^2 + y \text{ counts}^2 + z \text{ counts}^2)} \quad (3)$$

Here we show a summation of activity counts for daytime (07:00–22:59) and night-time (23:00–06:59) of each measurement day. Data were processed using ActiLife5 software (Actigraph, FL, USA).

## Continuous BP and HR: cuffless monitoring

BP and HR were recorded on a 24-h basis at B3, B1, D1, D3, D5 using SOMNOtouch™ NIBP system (SOMNOmedics, Germany). The system was placed at 8h:30–9h:30 and removed the following morning. Brachial blood pressure for calibration was measured with an Omron automatic blood pressure monitor. BP was determined with the Pulse Transit Time (PTT) continuously, using an ECG and the $SPO_2$ finger clip. Data were analyzed using Domino light 1.4 software (SOMNOmedics, Germany).

## Continuous core body temperature: ingestible pill telemetry

Telemetric system e-Celsius® Performance with e-Performance manager 1.3.2 software (BodyCap, France) was used for continuous gastrointestinal temperature monitoring. Single-use pills were activated, swallowed, and continuously recorded core body temperature twice per minute. Because the pill is eliminated naturally with the feces, a new pill was ingested every day from B3 to R1 (10 pills per participant). When more than one pill was present in digestive tract, gastrointestinal temperature was calculated as mean value for several pills. We calculated average daytime temperature (mean from 07:00 to 22:59), night-time temperature (mean from 23:00 to 06:59), and mesor temperature (mean for 24 h) for all measurement days. As the pills are not certified MRI-compatible, 6 out of 18 participants were assigned to the core body T° recording procedure, and did not perform MRI test.

## Glucose tolerance: OGTT and HOMA-IR

An OGTT was performed at B2 and D3 (48 h of immersion) in the morning. Plasma glucose and insulin were measured at baseline (fasting) and 30, 60, and 120 minutes after ingestion of 75 g of glucose. The total area under the curve (AUC) for both plasma glucose and insulin was calculated using the trapezoidal rule. Matsuda insulin sensitivity index (ISI) was calculated as follows (with glucose in mmol.L⁻¹ and insulin in μIU.mL⁻¹):

$$\text{Matsuda ISI} = \frac{10.000}{\sqrt{[\text{fasting glucose} \times 18 \times \text{fasting insulin}] \times [\text{mean glucose} \times 18 \times \text{mean insulin during OGTT}]}} \quad (4)$$

The homeostasis model assessment of insulin resistance was calculated at B1, D3, D5 as follows (with glucose in mmol.L⁻¹ and insulin in μIU.mL⁻¹):

$$\text{HOMA-IR} = \frac{\text{fasting insulin} \times \text{fasting glucose}}{22.5} \quad (5)$$

## Resting metabolic rate and non-protein respiratory quotient: indirect calorimetry

At B5, B1 and D5, oxygen consumption ($\dot{V}O_2$) and carbon dioxide excretion ($\dot{V}CO_2$) were measured. Two 20-minutes measurements were performed by canopy dilution respirometry (Quark RMR, Cosmed, Italy) using OMNIA 2.0 software (Cosmed, Italy). Post-measurement simulations were performed in a mixing chamber using a gas tank of 16.15% $O_2$ and 4.0% $CO_2$ to correct raw data of $\dot{V}O_2$ and $\dot{V}CO_2$. Participants were in fasting state, at rest, and thermo-neutrality. RMR was calculated using the Ferrannini equation[81], and non-protein respiratory quotient was calculated as the ratio between non-protein $\dot{V}CO_2$ and $\dot{V}O_2$.

## Nailfold capillary density: capillaroscopy

Nailfold capillaroscopy was performed on the middle finger of the right hand on B1, D4, and R1. Images were recorded using a capillary microscope Dino-Lite MEDL4N Pro with DinoCapture 2.0 software (Dino-Lite Medical, France). Capillary density was estimated by counting capillary number per mm.

## Statistical analysis

Data are presented as mean ± s.e.m for text and bar plots, and as minimum, 25th percentile, median, 75th percentile, and maximum for box plots, if not indicated otherwise. The effect of DI on the variables of interest was tested by repeated measures ANOVA followed by a Bonferroni post-hoc test to account for multiple comparisons. Relationships between variables of interest were examined using the Pearson correlation coefficient. Statistical significance was set at adjusted $P \le 0.05$. Statistical analyses were performed with GraphPad Prism 9.4.0.

## Reporting summary

Further information on research design is available in the Nature Portfolio Reporting Summary linked to this article.

# Data availability

Derived data supporting the findings of this study (individual dei-dentified dataset for the data presented in all the figures and tables of this paper, and data dictionary), as well as the entire protocol as submitted for regulatory approvals, are permanently available on request from the corresponding authors (A.R., M-A.C., N.N.) within a maximum response delay of one month. The scientific community can request access to the raw data resulting from ESA's Dry Immersion studies for retrospective studies. ESA is committed to making this data available in order to promote scientific progress and knowledge sharing. After data curation and handling, the data will be archived at the ESA HRE Data Archive, mandated by ESA's Human and Robotic Exploration (HRE) Programs Directorate (link here: https://hreda.esac.esa.int/hreda/#/pages/home). Data will be made available by ESA, i.e., after a request has been made and dully justified. The latter is in line with ESA's Personal Data Protection Framework rules (https://esamultimedia.esa.int/docs/LEX-L/ESA_Principles_of_PDP_Rules_of_Procedure_for_DPSA_and_Policy.pdf).

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

## Acknowledgements

We thank all the participants who volunteered for this dry immersion investigation. We thank the nurses, nutrition team, paramedical, and medical team at the MEDES Space Clinic (Toulouse Rangueil Hospital). We thank Pr. Patrick Saulnier from Angers University and Hospital for his contribution to statistical analysis. The VIVALDI study was funded by the European Space Agency (ESA), and the French National Center of Space Studies (Centre National d'Études Spatiales, CNES) has been the Promoter of the study according to French law. A. Robin received a Ph.D. grant from CNES and Région des Pays de la Loire. The protocol was also supported by specific CNES funding n° 4800001118. Schemas and artworks in Figs. 1b–d, 2g, 3d, 4d, 5a, d, 6a, b, 7d, g, and 9b were created with BioRender.com (full version).

## Author contributions

A.R., C.L., A.B., L.V., A.P.L.T., A. Choukér, D.A.G., M.T., A. Chopard, P.D., H.N., S.B., E.R., M.-A.C., and N.N. led the investigator team across the study (ESA expert group). A.V.O. and I.A. from ESA funded and designed the study. A.R. and N.N. performed and analyzed plasma volume assessment, bio-impedance tests, continuous recordings of heart rate, blood pressure, and core temperature, calf plethysmography, and capillaroscopy. L.V., M.-T.L., and P.F. performed and analyzed the bone biomarkers. M.K. performed and analyzed the ophthalmological measurements. A. Chopard and G.P. analyzed the muscle force parameters. C.L., I.D.G., D.L., I.H.-F., and C.S. performed and analyzed the indirect calorimetry. F.L. performed the main blood samples analysis. G.G.-K. from CNES represented the Promoter of the study. M.-P.B. and R.B.D.V. coordinated the study and experiments by the medical and paramedical teams at the MEDES clinic. A.R., M.-A.C., and N.N. drafted the manuscript. All authors reviewed, edited, and approved the manuscript.

## Competing interests

The authors declare no competing interests.

## Additional information

[1]Univ Angers, CRC, CHU Angers, Inserm, CNRS, MITOVASC, Equipe CARME, SFR ICAT, F-49000 Angers, France. [2]European Space Agency (ESA), Noordwijk, The Netherlands. [3]Institute of Metabolic and Cardiovascular Diseases, INSERM, Paul Sabatier University, UMR1297 Toulouse, France. [4]Anschutz Health and Wellness Center, Division of Endocrinology, University of Colorado, Aurora, CO, USA. [5]INSERM, University Jean Monnet, Mines Saint-Etienne, U 1059 Saint Etienne, France. [6]Department of Neurology, CHU Toulouse and I2MC-INSERM 1297, Toulouse, France. [7]DMEM, Montpellier University, INRAE, Montpellier, France. [8]Centre of Human and Applied Physiological Sciences, King's College London, London, UK. [9]Extreme Environments Laboratory, School of Sport, Health and Exercise Science, University of Portsmouth, Portsmouth PO1 2EF, UK. [10]Laboratory of Translational Research Stress and Immunity, Department of Anesthesiology, Hospital of the Ludwig-Maximilians-University (LUM), Munich, Germany. [11]Université de Caen Normandie, Inserm, COMETE U1075, CYCERON, CHU de Caen, F-14000 Caen, France. [12]DEPE-IPHC - Département Ecologie, Physiologie et Ethologie, Strasbourg, France. [13]CarMeN Laboratory, INSERM 1060, INRA 1397, University Claude Bernard Lyon1, Human Nutrition Research Center Rhône-Alpes, Oullins, France. [14]Faculty of Sport Sciences, Université de Reims Champagne-Ardenne, Reims, France. [15]Laboratoire de Biochimie, CHU d'Angers, Angers, France. [16]Telespazio Belgium S.R.L. for the European Space Agency, Noordwijk, The Netherlands. [17]Centre National d'Études Spatiales (CNES), Paris, France. [18]Institute of Space Physiology and Medicine (MEDES), Toulouse, France. ✉e-mail: adrirobin@etud.univ-angers.fr; macustaud@chu-angers.fr; nastassia.navasiolava@chu-angers.fr

