## [Peer Review File · Nature Communications]

Comprehensive assessment of physiological responses in women during the ESA dry immersion VIVALDI microgravity simulationREVIEWER COMMENTS

Reviewer #1 (Remarks to the Author):

This is a well-designed and reported study and very important.

- One key issue that I had with this paper is that it intends to develop baseline data for future studies, so it doesn't make sense to draw generalizable conclusions. However, it would be useful to include a section in the discussion part that suggests how future researchers should design studies in this field and utilize this data for comparison, while also considering potential biases. For example, you can explain how and what participants to recruit in future studies to be comparable, you could discuss how the recruitment strategy and the selection of individual volunteers could introduce a selection bias in future comparative studies. Additionally, you could address the issue that due to the limited sample size and the absence of a comparison group, some of the measurements might be reflective of random error rather than a clear indication of changes specifically caused by dry immersion. There is two distinct sections in the paper (it is clearer in the confidential protocol and would be useful it be clear in the paper) whether Dry immersion is feasible to be conducted in women and If yes how. So be useful if there is a paragraph before global tolerance on feasibility on running DI for women (which I assume its feasible) and what adjustments made it possible to be feasible. In discussion, It will be useful, there will be a section how your adjustment for DI was similar or different from the Russia DI study on six women.
- I assume that there was no blinding in the outcome assessment, and the individuals who assessed the patients were aware of their conditions. It would be useful to explicitly state whether blinding was not performed or utilized, if applicable.
- In the abstract, you mention "Slight sex differences are revealed in...". Since you did not include men as a comparison group in your study, I suggest either removing that sentence or clarifying the comparative dataset on sex differences, such as known data from other dry immersions or space missions..
- Regarding the sentence, "To determine the sample size, we based it on combined considerations of physiological assessments, taking into account already known effects of DI," could you provide further explanation? Which physiological assessments were used to calculate the sample size, and what approach was taken in the calculation process?

- “One subject left the protocol on the first day of immersion due to technical issues, and another was excluded 121 before the start of immersion for regulatory reasons related to clinical trials.” Could you explain this further, at the moment, it is not clear what regulatory reason for clinical trials or technical issues means?
- As this paper is a baseline framework for future studies. It be useful on page 12 that you mention the mean temperature data in the text in this section (it is difficult to extract the data accurately from the figure) . “The baseline curve represents 459 the mean for days from B3 to B1, during DI - from D1 to D5, and recovery - from R0 to R1. We also calculated a daytime 460 (07:00 – 22:59), a nighttime (23:00 – 06:59), and a mesor (24h) mean temperature for each day (Fig. 5e). During DI, core 461 body temperature decreased of about 0.15°C, while circadian rhythms were globally preserved”.
- It would be useful to add a sentence in discussion on a potential different confounder e.g. stress or boredom form the situation and loneliness especially for those who were on dry immersion alone and its impact on the data.
- The image on page 29 of the protocol MEDES 20-232 is quite helpful and would be useful if it is part of the article.
- Do the authors consider to put the protocol / experimental protocol available online and/or dataset if the paper is accepted? It be useful if that’s possible as there are a lot of details that are quite helpful in replicating the study or using the data.
- There was a series of data that was reported as being collected in protocol but I couldn’t find it in the main article like heparan sulfate, zonulin or some of the proteomics to understand the blood clot formation. Apologies if I missed them but if not, please report what happened to those data as otherwise it can introduce selective outcome reporting bias.
- In this sentence, provide a reference where the male data come from - “Slight sex differences are revealed in response to DI, such as smaller increase in ONSD (10-15% vs 20% in men), more pronounced orthostatic intolerance (41% decrease in OT-index vs 31% in men), and somewhat greater impairment in glucose metabolism (41% decrease in Matsuda index vs 31% in men).”

Reviewer #2 (Remarks to the Author):

Manuscript by Robin A. et al. is an excellent example of a comprehensive study of the response of various human physiological systems under simulated weightlessness. The authors of the work have a strong background in conducting model experiments and working with the techniques described in the article. The protocol of the experiment and all methods are described very correctly and absolutely clearly. The presented results are described clearly, the graphic material fully reflects the data obtained. I believe that all these results may be of interest to a wide circle of readers.

However, this study is not free from some shortcomings.

The study included female subjects, but the onset of dry immersion exposure was not synchronized to any phase of the menstrual cycle. Moreover, at the selection stage, there was a wide range of subjects by means of contraception (line 113-114) and, consequently, by the state of the reproductive system. This is a limitation of this study and may have an impact on the metabolic parameters that are described in the article.

The literature presents data on the effect of 5-day dry immersion on the female reproductive system (Gorbacheva EY, Toniyan KA, Biriukova YA, Lukicheva NA, Orlov OI, Boyarintsev VV, Ogneva IV. The State of the Organs of the Female Reproductive System after a 5 -Day "Dry" Immersion *Int J Mol Sci* 2023 Feb 19;24(4):4160 doi:

10.3390/ijms24044160 PMID: 36835572) which shows a sharp decrease in progesterone after immersion. This may well be due, for example, to a lower intake of cholesterol in tissues and, as a result, its accumulation in the blood. These data should be discussed.

The statement that the effects of dry immersion are superior to those of the menstrual cycle (lines 524-528) seems too strong in this context and should be made more cautious.

As a minor remark, the authors should explain what caused the choice of exposure duration, since 5 days does not correlate with the current duration of space flights.

Point by point Response to reviewers' comments

Dear reviewers,

Many thanks for your time and your attention to our work, and for such a positive appreciation and review. We greatly appreciate all your comments, and we improved the paper accordingly.

Please find below our point-by-point answers.

We also would like to add Chantal Simon (from the Human Nutrition Research Center of the University Claude Bernard, Lyon, France) as co-author for the work she has done in calorimetry data treatment. In an effort to homogenize presentation of the data in the spaceflight-analog studies conducted by our teams, RMR calculated from the equation of Ferrannini relying on substrates utilization rather than on Weir's equation is now presented. Results are only marginally different (less than 2%) and do not change the overall message on metabolic data

We'll be very grateful if you can consider this request.

Also, the abstract has been rewritten to fit within the 150 words limit.

To make questions and answers easily detected:

- Reviewers' comments are here in black,
- Our answers are in blue,
- Manuscript initial text is in grey,
- Manuscript modifications are in green.

Reviewer #1 (Remarks to the Author):

Reviewer 1: This is a well-designed and reported study and very important.

We deeply appreciate this positive and supportive comment

Reviewer 1: One key issue that I had with this paper is that it intends to develop baseline data for future studies, so it doesn't make sense to draw generalizable conclusions. However it would be useful to include a section in the discussion part that suggests how future researchers should design studies in this field and utilize this data for comparison, while also considering potential biases. For example, you can explain how and what participants to recruit in future studies to be comparable, you could discuss how the recruitment strategy and the selection of individual volunteers could introduce a selection bias in future comparative studies.

Answer: We agree with the reviewer, we add to the Discussion a specific paragraph (p.19, l.778-793):

Comparability and bias considerations for recruitments in future research

To improve comparability of obtained data with future studies, we might consider recruitment of participants within similar ranges of age, fitness ($\dot{V}O_{2max}$), anthropometry (height, BMI), naïve to DI and preferably also with no previous participation to any spaceflight analogs. To limit potential bias, it

would be also useful for future studies to better characterize the subjects before experiment, not only via questionnaires/interview as done in the present study, but also via measurement of their usual daily lifestyle, i.e., physical activity level by actigraphy, food habits by dietary survey.

Reviewer 1: Additionally, you could address the issue that due to the limited sample size and the absence of a comparison group, some of the measurements might be reflective of random error rather than a clear indication of changes specifically caused by dry immersion.

Answer: We agree with the reviewer and we add to the paragraph “Limitations” (p.19, l.762-765):

To our knowledge, 18 participants is the largest sample size ever used for a single DI protocol without countermeasures, though it remains limited. Even if random error for some measurements cannot be excluded, in this study we were able to identify significant changes for major functions with underlying physiological explanation or hypotheses. Individual data points are shown to give the reader an exact idea on inter-individual variability.

Reviewer 1: There is two distinct sections in the paper (it is clearer in the confidential protocol and would be useful it be clear in the paper) whether Dry immersion is feasible to be conducted in women and If yes how. So be useful if there is a paragraph before global tolerance on feasibility on running DI for women (which I assume it's feasible) and what adjustments made it possible to be feasible.

Answer: We agree with the reviewer and add to the Results (p.9, l.352-360):

Feasibility and adjustments for DI on women

The 5-day DI without authorized periods of sitting or standing was proved feasible in women. No major medical issues or adverse events were reported. For daily shower and weighting, participants were transferred to a specific platform maintained at -6° head-down position. To limit discontinuation of immersion for urination and perform urine collection, Purewick™ system was available. This system, primarily designed for women suffering from urinary incontinence, aspirates urine from a soft flexible wick (external catheter) to a sealed collector. Participants positioned the system by themselves before urination, then removed until the next urination. Also, bed-side urinary basin for urination in supine position out-of-bath on lifting platform was available. Preferences and requests of subjects were respected. Three participants used only Purewick, three used both options (mostly Purewick), and twelve used only basin.

Also, we add to Global tolerance and out-of-bath time (p.9, l.373-374):

... Out-of-bath time was about 10 h for a total of 120 h, and no more than 3 h per day (Supplementary Data Fig. 1c), which is comparable to the usual out-of-bath time for DI protocols. For example, out-of-bath time during the 5-day DI protocol conducted in 18 men (MEDES, France, 2018-2019) was $9.7 \pm 1.3 \text{h}^{17}$. Urination option had only minor effect on total out-of-bath time, which consisted 9h 16min for Purewick only, 9h 19 min for combined, and 10h 47 min for basin only (Supplementary Data Fig. 1d).

We add to the Supplementary Fig.1 the following table in part d describing the urination options, the related time spent in bath, and the number of participants who used each method:

Reviewer 1: In discussion, It will be useful, there will be a section how your adjustment for DI was similar or different from the Russia DI study on six women.

Answer: We agree with the reviewer and we add to the Discussion (p.19, l.750-759):
Methodological comparison with Naiad-2020 study

A first DI study in women of reproductive age, Naiad-2020, was conducted in Russia in September-November 2020⁹⁻¹¹. This pilot DI lasted for 3 days and included 6 healthy women of 24-39 yrs. Special “home-made” universal portable device for mechanical urine collection in women was designed (for details see Tomilovskaya et al., 2020⁹), allowing for most cases urination in bath (with a pillow under the back and shoulders), though sometimes a solid support was needed. During immersion, subjects spent 20-30 min.day⁻¹ out-of-bath, including 7-9 min.day⁻¹ sitting and standing (“non-strict” DI protocol). Regarding to menstrual cycle, the onset of Naiad DI was standardized – all 6 participants began DI at the 7th day of the cycle and completed it at the 10th day. Water intake was ad libitum. Diet was standardized from B1 to R0 with caloric intake of 2200-2300 kcal.day⁻¹; taste preferences of the participants were taken into account, and some adaptations from the proposed menu were allowed to maintain a comfortable psychophysiological state during DI.

Reviewer 1: I assume that there was no blinding in the outcome assessment, and the individuals who assessed the patients were aware of their conditions. It would be useful to explicitly state whether blinding was not performed days or utilized, if applicable.

Answer: We agree with the reviewer and we add to the Methods in the paragraph “General protocol” (p.4, l.158-159):
 This protocol was designed as Single-group assignment and Open label study (no blinding was imposed by protocol for data collection and analyses).

Reviewer 1: In the abstract, you mention "Slight sex differences are revealed in...". Since you did not include men as a comparison group in your study, I suggest either removing that

sentence or clarifying the comparative dataset on sex differences, such as known data from other dry immersions or space missions.

Answer: We agree with your comment. We had to shorten the abstract to comply with authorized length of 150 words, but we add to the Discussion in the paragraph “Conclusion and perspectives” (p.20, l.793-798):

Compared to known data from dry immersion studies with male participants, only slight sex differences are revealed in response to DI, such as smaller increase in ONSD (10-15% vs 20% in men – data from a 3-day DI³³ and from a 5-day DI³²), more pronounced orthostatic intolerance (41% decrease in OT-index vs 31% in men – recalculated from personal data for control group of a 5-day DI¹⁷, to allow comparison for this parameter), and somewhat greater impairment in glucose metabolism (41% decrease in Matsuda index vs 31% in men – recalculated from personal data of a 3-day DI⁷¹, to allow comparison for this parameter).

With references:

17. Robin, A. et al. DI-5-CUFFS: Venorestrictive Thigh Cuffs Limit Body Fluid Changes but Not Orthostatic Intolerance Induced by a 5-Day Dry Immersion. *Front. Physiol.* 11, 383 (2020).

32. Kermorgant, M. et al. Effects of Venorestrictive Thigh Cuffs on Dry Immersion-Induced Ophthalmological Changes. *Front. Physiol.* 12, 692361 (2021).

33. Kermorgant, M. et al. Impacts of Simulated Weightlessness by Dry Immersion on Optic Nerve Sheath Diameter and Cerebral Autoregulation. *Front. Physiol.* 8, 780 (2017).

71. De Abreu, S. et al. Multi-System Deconditioning in 3-Day Dry Immersion without Daily Raise. *Front. Physiol.* 8, 799 (2017).

Reviewer 1: Regarding the sentence, "To determine the sample size, we based it on combined considerations of physiological assessments, taking into account already known effects of DI," could you provide further explanation? Which physiological assessments were used to calculate the sample size, and what approach was taken in the calculation process?

Answer: We agree with the reviewer and we modify the Methods section in the paragraph “Participants” (p.3, l.108-114):

Participants

Power-based calculation of the number of subjects is not directly applicable for such explorative studies, so to determine sample size, we based on combined considerations of physiological assessments, taking into account already known effects of DI. Thus, taken into account data obtained with previous MEDES DI in men, minimal sample to expect statistically significant difference between Pre- and Post-DI (at power 80% and alpha level 0.05), could be estimated as n=5 for plasma volume evolution (effect size 2.01), n=11 for orthostatic tolerance (effect size 0.96), n=15 for glucose tolerance (effect size 0.8), n=19 for $\dot{V}O_{2max}$ (effect size 0.69). Therefore, a total of 20 subjects was deemed necessary for the study and approved by the national Ethic Committee.

Reviewer 1: “One subject left the protocol on the first day of immersion due to technical issues, and another was excluded before the start of immersion for regulatory reasons related to clinical trials.” Could you explain this further, at the moment, it is not clear what regulatory reason for clinical trials or technical issues means?

Answer: We agree with the reviewer and we add to the Methods in the paragraph “Participants” (p.3, l.124-128):

Twenty healthy women were recruited. All of them were informed about the experimental procedures and gave their written consent. One participant left the protocol on the first day of immersion due to technical issue (technical problem with bath lifting platform which could not be fixed quickly without emptying the bath, leading to the decision to stop the study for this participant), and another could not

be included for regulatory reasons related to clinical trials (this participant unexpectedly was still during the exclusion period following participation in a previous unrelated clinical trial). They were excluded from analysis.

Reviewer 1: As this paper is a baseline framework for future studies. It be useful on page 12 that you mention the mean temperature data in the text in this section (it is difficult to extract the data accurately from the figure). “The baseline curve represents the mean for days from B3 to B1, during DI - from D1 to D5, and recovery - from R0 to R1. We also calculated a daytime (07:00 – 22:59), a nighttime (23:00 – 06:59), and a mesor (24h) mean temperature for each day (Fig. 5e). During DI, core body temperature decreased of about 0.15°C, while circadian rhythms were globally preserved”.

Answer: We agree with the reviewer and add to the Results (p.12, l.483-485):

Temperature data averaged over 15 min are plotted in Fig. 5d. The baseline curve represents the mean for days from B3 to B1, during DI - from D1 to D5, and recovery - from R0 to R1. We also calculated a daytime (07:00 – 22:59; $37.18 \pm 0.04^\circ\text{C}$ at B vs. 37.08 ± 0.07 at DI vs. 37.31 ± 0.05 at R), a nighttime (23:00 – 06:59; 36.72 ± 0.07 at B vs. 36.57 ± 0.08 at DI vs. 36.81 ± 0.07 at R), and a mesor (24h) mean temperature (37.02 ± 0.05 at B vs. 36.90 ± 0.07 at DI vs. 37.15 ± 0.06 at R) for each day (Fig. 5e). During DI, core body temperature decreased of about 0.10 - 0.15°C, while circadian rhythms were globally preserved.

Reviewer 1: It would be useful to add a sentence in discussion on a potential different confounder e.g. stress or boredom form the situation and loneliness especially for those who were on dry immersion alone and its impact on the data.

Answer: We agree with the reviewer and we add to the Discussion in the paragraph “Limitations” (p.19, l.766-770):

Various confounders could potentially impact results under DI, such as individual predisposition to stress and boredom, despite high motivation of all the subjects to do their best. Loneliness could also appear, especially for two subjects without room-mates who underwent DI alone. The staff was particularly careful to these subjects and spent more time with them, being more available with twice less workload. They were not different from the rest of the group in discomfort, sleep quality, and 24-h cortisol.

Reviewer 1: The image on page 29 of the protocol MEDES 20-232 is quite helpful and would be useful if it is part of the article.

Answer: We agree with the reviewer and we modified the Fig. 1d, adding the notion of selection visit and some iconography as mentioned in the image of the protocol p.29:

Fig.1 | The DI model in MEDES space clinic (Toulouse, France).

a, Two participants in two separate baths underwent strict DI simultaneously. **b,c,** The DI model simulates the lack of support and unloading observed in microgravity, as shown by the superficial pressure mapping where the pressure applied by the body is evenly distributed over the sensors during bath immersion. **d,** Operational scenario and main testing protocols are represented through the study period. The participants arrived at the facility in the evening of B5, four days of baseline data collection have been performed (B4-B1). The participants entered into the bath tubs at D1 around 09:30 and left the bath at the same hour at R0 (five full days of DI: D1-D5). Two days of recovery data collection were performed (R0-R1). Participants left the facilities in the morning of R2. The tests performed are represented by a filled dot, and a solid line indicates a continuous recording.

Reviewer 1: Do the authors consider to put the protocol / experimental protocol available online and/or dataset if the paper is accepted? It be useful if that's possible as there are a lot of details that are quite helpful in replicating the study or using the data.

Answer: We add to the Data Availability Statement section (p.21, l.835-840):

DATA AVAILABILITY STATEMENT

The scientific community can request access to the raw data resulting from ESA's Dry Immersion studies for retrospective studies. ESA is committed to making this data available in order to promote scientific progress and knowledge sharing.

After data curation and handling, the data will be archived at the ESA HRE Data Archive, mandated by ESA's Human and Robotic Exploration (HRE) Programs Directorate (link here: <https://hreda.esac.esa.int/hreda/#/pages/home>). Data will be made available by ESA, i.e., after a request has been made and dully justified. The latter is in line with ESA's Personal Data Protection Framework rules (https://esamultimedia.esa.int/docs/LEX-L/ESA_Principles_of_PDP_Rules_of_Procedure_for_DPDA_and_Policy.pdf). The entire protocol, as submitted for regulatory approvals, will be available too on the ESA HRE Data Archive website.

Reviewer 1: There was a series of data that was reported as being collected in protocol but I couldn't find it in the main article like heparan sulfate, zonulin or some of the proteomics to

understand the blood clot formation. Apologies if I missed them but if not, please report what happened to those data as otherwise it can introduce selective outcome reporting bias.

Answer: This 5-day strict DI in women was an integrative international study promoted by ESA and CNES. The major part of data yielded from this protocol was managed by the expert group and analyzed in this paper. However, there were also few ancillary studies for other dedicated teams, such as “redox balance and inflammatory biomarkers”, “biological vascular parameters”, “coagulation and thrombosis”, “brain MRI”, which will be published later separately.

Reviewer 1: In this sentence, provide a reference where the male data come from - “Slight sex differences are revealed in response to DI, such as smaller increase in ONSD (10-15% vs 20% in men), more pronounced orthostatic intolerance (41% decrease in OT-index vs 31% in men), and somewhat greater impairment in glucose metabolism (41% decrease in Matsuda index vs 31% in men).”

Answer: Thank you for your comment, as mentioned above we add the following explanations and references (p.20, l.793-798):

Compared to known data from dry immersion studies with male participants, only slight sex differences are revealed in response to DI, such as smaller increase in ONSD (10-15% vs 20% in men – data from a 3-day DI³³ and from a 5-day DI³²), more pronounced orthostatic intolerance (41% decrease in OT-index vs 31% in men – recalculated from personal data for control group of a 5-day DI¹⁷, to allow comparison by this parameter), and somewhat greater impairment in glucose metabolism (41% decrease in Matsuda index vs 31% in men – recalculated from personal data of a 3-day DI⁷¹, to allow comparison by this parameter).

With references:

17. Robin, A. et al. DI-5-CUFFS: Venopressive Thigh Cuffs Limit Body Fluid Changes but Not Orthostatic Intolerance Induced by a 5-Day Dry Immersion. *Front. Physiol.* 11, 383 (2020).

32. Kermorgant, M. et al. Effects of Venopressive Thigh Cuffs on Dry Immersion-Induced Ophthalmological Changes. *Front. Physiol.* 12, 692361 (2021).

33. Kermorgant, M. et al. Impacts of Simulated Weightlessness by Dry Immersion on Optic Nerve Sheath Diameter and Cerebral Autoregulation. *Front. Physiol.* 8, 780 (2017).

71. De Abreu, S. et al. Multi-System Deconditioning in 3-Day Dry Immersion without Daily Raise. *Front. Physiol.* 8, 799 (2017).

Reviewer #2 (Remarks to the Author):

Reviewer 2: Manuscript by Robin A. et al. is an excellent example of a comprehensive study of the response of various human physiological systems under simulated weightlessness. The authors of the work have a strong background in conducting model experiments and working with the techniques described in the article. The protocol of the experiment and all methods are described very correctly and absolutely clearly. The presented results are described clearly, the graphic material fully reflects the data obtained. I believe that all these results may be of interest to a wide circle of readers.

We thank the reviewer for these very kind words

Reviewer 2: However, this study is not free from some shortcomings.

The study included female subjects, but the onset of dry immersion exposure was not synchronized to any phase of the menstrual cycle. Moreover, at the selection stage, there was a wide range of subjects by means of contraception (line 113-114) and, consequently, by the

state of the reproductive system. This is a limitation of this study and may have an impact on the metabolic parameters that are described in the article.

The literature presents data on the effect of 5-day dry immersion on the female reproductive system (Gorbacheva EY, Toniyan KA, Biriukova YA, Lukicheva NA, Orlov OI, Boyarintsev VV, Ogneva IV. The State of the Organs of the Female Reproductive System after a 5 -Day "Dry" Immersion *Int J Mol Sci* 2023 Feb 19;24(4):4160 doi: 10.3390/ijms24044160 PMID: 36835572) which shows a sharp decrease in progesterone after immersion. This may well be due, for example, to a lower intake of cholesterol in tissues and, as a result, its accumulation in the blood. These data should be discussed.

Answer: We thank the reviewer for this constructive point! We slightly improved the Discussion with the nutrition team, and add to the paragraph "Metabolic alterations" (p.18, l.710-723):

Metabolic alterations

Loss of lean body mass is a systematic observation in response to simulated and actual microgravity that is proportional to the duration of the exposure⁶⁷. Five days of dry immersion induced pronounced deconditioning in women, as it was sufficient to reduce fat-free mass and trigger greater protein catabolism. It was further associated with the development of metabolic alterations including the onset of metabolic stress, dyslipidemia, and decreases in insulin sensitivity and glucose control. Prior HDBR and in-flight studies showed that the organism perceives weightlessness as a stress. During Spacelab Life Sciences-1 (SLS1) and SLS2 missions, the whole protein turnover, i.e., both protein synthesis and degradation⁶⁸, acute phase hepatic proteins, cortisol secretion, and biomarkers of systemic inflammation (IL-1 and IL-10) increased during the first day of flight. Medium-term HDBR (42 days) reported increases in urinary excretion of cortisol, urea, creatinine, and growth hormone (GH) to last around 4-5 weeks⁶⁹. Decreases in insulin sensitivity were reported as soon as after 3 days of HDBR⁷⁰ and DI in male adults⁷¹. Indeed, three days of strict DI increased HOMA-IR by 43±11% and Net insulin response by 72±23%, and decreased Matsuda ISI by 31±7%⁷¹. This reduced insulin sensitivity is likely associated with a blunted insulin-stimulated glucose disposal as shown recently after 3 days of HDBR in healthy males⁷². We observed no shift in the use at fast of substrate from lipid toward carbohydrate oxidation after 5 days of DI. Alterations in substrate metabolism may take longer to be apparent. Indeed, while 3 days of HDBR failed to detect changes in insulin-stimulated carbohydrate and fat oxidation, changes in fuel selection were reported after 7 days of HDBR⁷³ and after the long term HDBR^{74,75} in both male and female adults. In agreement with results obtained in long-term HDBR⁷⁵, pronounced increases in fasting plasma TG, LDL, NEFA, and AIP were finally observed suggesting atherogenic effects of 5 days of DI. This raises a question of cardiovascular risk for long-term missions. Of note, while sharp decreases in plasma progesterone concentration were reported after 5 days of DI compared to ambulatory conditions on the same day of the cycle in healthy females⁷⁶, menstrual cycle was not controlled in our female participants. Although this limitation needs to be acknowledged, we did not observe differential metabolic responses to DI between individuals with high versus low plasma progesterone and estradiol levels, suggesting that the effects of the DI may be stronger than hormonal modifications. In addition, a drop in progesterone would be expected to counteract the metabolic effects of DI⁷⁷.

With references:

67. Adams, G. R., Caiozzo, V. J. & Baldwin, K. M. Skeletal muscle unweighting: spaceflight and ground-based models. *J Appl Physiol* (1985) 95, 2185–2201 (2003).

68. Stein, T. P., Leskiw, M. J. & Schluter, M. D. Diet and nitrogen metabolism during spaceflight on the shuttle. *J Appl Physiol* (1985) 81, 82–97 (1996).

69. Blanc, S. et al. Energy and water metabolism, body composition, and hormonal changes induced by 42 days of enforced inactivity and simulated weightlessness. *J Clin Endocrinol Metab* 83, 4289–4297 (1998).

70. Bergouignan, A., Rudwill, F., Simon, C. & Blanc, S. Physical inactivity as the culprit of metabolic inflexibility: evidence from bed-rest studies. *J Appl Physiol* (1985) 111, 1201–1210 (2011).

71. De Abreu, S. et al. Multi-System Deconditioning in 3-Day Dry Immersion without Daily Raise. *Front. Physiol.* 8, 799 (2017).

72. Shur, N. F. et al. Human adaptation to immobilization: Novel insights of impacts on glucose disposal and fuel utilization. *J Cachexia Sarcopenia Muscle* 13, 2999–3013 (2022).

73. Blanc, S. et al. Fuel homeostasis during physical inactivity induced by bed rest. *J Clin Endocrinol Metab* 85, 2223–2233 (2000).

74. Bergouignan, A. et al. Physical inactivity differentially alters dietary oleate and palmitate trafficking. *Diabetes* 58, 367–376 (2009).
75. Bergouignan, A. et al. Effect of physical inactivity on the oxidation of saturated and monounsaturated dietary Fatty acids: results of a randomized trial. *PLoS Clin Trials* 1, e27 (2006).
76. Gorbacheva, E. Y. et al. The State of the Organs of the Female Reproductive System after a 5-Day ‘Dry’ Immersion. *Int J Mol Sci* 24, 4160 (2023).
77. Kalkhoff, R. K. Metabolic effects of progesterone. *Am J Obstet Gynecol* 142, 735–738 (1982).

We also add to the Discussion in the paragraph “Limitations” (p.19, l.771-776):

Another limitation of our study is that the onset of DI was not synchronized with menstrual cycle phase. Individual cycle phases respective to protocol were deduced from sex hormones profiles. As already stated, our data did not show a significant impact of the menstrual cycle on DI effects, but this impact remains possible. Furthermore, the question of the effect of DI on cycle and more generally on reproductive system was not explored in our study. Interestingly the very recent data on the effect of 5-day dry immersion on the female reproductive system⁷⁶ show a sharp decrease in progesterone after immersion.

With references:

76. Gorbacheva, E. Y. et al. The State of the Organs of the Female Reproductive System after a 5-Day ‘Dry’ Immersion. *Int J Mol Sci* 24, 4160 (2023).

Reviewer 2: The statement that the effects of dry immersion are superior to those of the menstrual cycle (lines 524-528) seems too strong in this context and should be made more cautious.

Answer: We agree with the reviewer and modify the Results (p.14, l.549-554):

DI effects override and menstrual cycle effects

Cycle phase was deduced from blood progesterone and estradiol dynamics. Individual cycle phases respective to protocol are indicated in Supplementary data Fig. 3. ~~DI effect gains over cycle effect.~~ Our data do not show a significant impact of the menstrual cycle phase on the changes induced by DI. Participants with maximal blood progesterone during 5-day DI $>2\mu\text{g/L}$ set as follicular-to-luteal phase threshold (n=6) were not different in responses to DI. Similarly, those with maximal blood estrogen during 5-day DI $\geq 200\text{ ng/L}$ (n=7) did not differ.

Reviewer 2: As a minor remark, the authors should explain what caused the choice of exposure duration, since 5 days does not correlate with the current duration of space flights.

Answer: This is a very good point. To clarify our choice of 5 days, we add to the Methods in the paragraph “General protocol” (p.4, l.136-139):

We were interested in the early phase of adaptation to microgravity (corresponding to the initial phase of flight). Furthermore, we consider that towards D3-D4 of DI the new steady state, i.e., equilibrium is globally established following the acute transition (though for some systems this establishing is ongoing). As a result, a 5-day DI appears to be the best compromise to study this early phase.

REVIEWERS' COMMENTS

Reviewer #1 (Remarks to the Author):

The authors have done a great job in addressing comments and am happy with the article.

Reviewer #2 (Remarks to the Author):

Many thanks to the authors of the study, who paid attention to my comments and suggestions. I am completely satisfied with the answers and the changes made. I believe that the article has been significantly improved and can be accepted for publication.